# Improving Prompt-based Continual Learning with Key-Query Orthogonal Projection and Prototype-based One-Versus-All

## Abstract

Drawing inspiration from prompt tuning techniques applied to Large Language Models, recent methods based on pre-trained ViT networks have achieved remarkable results in the field of Continual Learning. Specifically, these approaches propose to maintain a set of prompts and allocate a subset of them to learn each task using a key-query matching strategy. However, they may encounter limitations when lacking control over the correlations between old task queries and keys of future tasks, the shift of features in the latent space, and the relative separation of latent vectors learned in independent tasks. In this work, we introduce a novel key-query learning strategy based on orthogonal projection, inspired by model-agnostic meta-learning, to enhance prompt matching efficiency and address the challenge of shifting features. Furthermore, we introduce a One-Versus-All (OVA) prototype-based component that enhances the classification head distinction. Experimental results on benchmark datasets demonstrate that our method empowers the model to achieve results surpassing those of current state-of-the-art approaches by a large margin of up to 20%. Our code is available at `https://anonymous.4open.science/r/KOPPA/README.md`.

## 1 Introduction

Continual Learning (CL) is an evolving field in machine learning, aiming to enable models to learn continuously from a sequence of tasks with varying data distributions. A challenging CL scenario is Class Incremental Learning (CIL), where a model sequentially learns new categories and must classify all seen classes without task-ID information, leading to a fundamental issue in CL known as Catastrophic Forgetting (CF) (French, 1999), where performance on earlier tasks degrades due to the absence of old task data and differences in data distributions.

In CIL, models are required to classify test samples without prior knowledge of their task IDs. Replay-based methods have shown promise in addressing this challenge (Buzzega et al., 2020; Gong et al., 2022; Guo et al., 2022) by using a buffer for old data. However, their performance still falls behind joint training, which combines all data for training from scratch. Storing more old data can boost performance, but becomes computationally expensive with a large number of tasks and raises privacy concerns. Recent approaches (Smith et al., 2023; Wang et al., 2022a;c) based on prompt tuning with large pretrained networks offer a compelling solution, achieving high performance without storing raw data and offering scalability.

Prompt-tuning methods adapt large pre-trained models to downstream tasks by learning prompts, achieving state-of-the-art data-free CIL results and scalability. The main ideas of those prompt-tuning approaches include storing prompts, using subsets for task learning, and employing query-key matching for inference. However, one of the key limitations of current prompt selection methods is that they suffer from the *mismatch in prompt representation between training and testing* and *feature shifting* during inference. To address this issue, we first propose a novel key-query training strategy inspired by MAML (Finn et al., 2017) and based on orthogonal projection. This approach aims to align new task keys perpendicularly to old task queries and therefore, not only reducing interaction between them but also preserving old task data representations.

Furthermore, we point out that the mismatch in prompt representation between training and testing examples further leads to the problem of inaccurately triggering the task classification head of an example. To address this problem, we adopt a prototype-based One-Versus-All (OVA) component to boost the task classification head distinction. Two components *key-query orthogonal projection* and *prototype-based OVA* are complementary to form our approach named **K**ey-query **O**rthogonal **P**rojection with **P**rototype-based OV**A** Continual Learning (KOPPA). To sum up, our contributions are as follows:

- We propose a novel training strategy inspired by MAML (Finn et al., 2017), which ensures an almost certain perpendicular constraint between future keys and past task queries, effectively eliminating feature shift.

- We propose a prototype-based OVA component to enhance the task classification head distinction.

- Experimental results on benchmark datasets not only demonstrate the effectiveness of our method in avoiding feature shifts but also show our state-of-the-art performance, exceeding the baselines up to a margin of 20%.

## 2 RELATED WORK

**Continual Learning:** Continual Learning (CL) methods are categorized into three groups (Van de Ven & Tolias, 2019). *Regularization-based methods* add constraints to parameter updates (Zenke et al., 2017; Aljundi et al., 2018) or enforce consistency with previous model outputs (Li & Hoiem, 2017; Zhu et al., 2021b). *Architecture-based methods* allocate parameter subsets and expand networks as needed, often requiring task IDs at evaluation (Mallya & Lazebnik, 2018; Wortsman et al., 2020). *Memory-based methods* achieve strong CL results by storing limited old data (Buzzega et al., 2020; Yoon et al., 2021) but face privacy issues. A recent approach gaining traction is *prompt tuning on large pre-trained networks* (Wang et al., 2022c;a; Smith et al., 2023), offering superior results without old data replay or task IDs. Our work focuses on this approach, addressing feature shift and improving traditional classification head quality.

**Prompt Tuning in CL:** Inspired by prompt-tuning's success in adapting vision transformer models (Huang et al., 2022; Zhou et al., 2022), a prompt-based CL approach is introduced. It competes effectively in CL settings without raw data storage, e.g., L2P (Wang et al., 2022c) and DualPrompt (Wang et al., 2022a). However, these methods may forget prompts and keys. In contrast, CODA (Smith et al., 2023) avoids forgetting by adding new prompts, keys, and masks for each new task, while freezing previous ones. Since CODA combines information of these prompts, keys and masks, from all tasks during evaluation, it can alter features of old samples compared to training.

**Feature drift and Classifier separation in CL:** Several works address feature drift in CL using supervised contrastive losses (Mai et al., 2021; Zhu et al., 2021b), known for their resistance to forgetting (Davari et al., 2022). Others improve CL methods by heuristically updating old feature positions. For example, Yu et al. (2020) estimates drift by comparing new data features before and after learning, while Joseph et al. (2022) uses an energy-based model for optimal feature regions in old data. Additionally, to enhance classification between old and new classes, it is suggested to incrementally store a buffer of old features or (mean, covariance) pairs for new classes introduced in each task, which are replayed with current data for balanced training (Zhu et al., 2021b;a). However, these methods assume feature stability in old data. Our work introduces a solution to prevent feature shifts, facilitating task differentiation without the need for old raw data replay.

## 3 OUR PROPOSED APPROACH

### 3.1 PROBLEM SETUP AND NOTATIONS

We consider the continual learning setting in which a model needs to learn from a sequence of classification tasks. Each task $t \in \{1, ..., T\}$ has a training dataset $D^t$ with $n_t$ i.i.d samples, and $T$ may be unknown or infinite. The model needs to learn these tasks sequentially, without replaying data from old tasks during the training stage, and without the task ID information during inference.

This is often known as *rehearsal-free Continual Learning* (Smith et al., 2022; Wang et al., 2022b; 2023a)

In this work, we propose solutions based on prompt tuning with pre-trained ViT (Dosovitskiy et al., 2021). We design our model as a composition of two components: *a pre-trained ViT backbone* $f_\Phi$ and *a classification head* $h_\theta$. Similar to other prompt-tuning works, we incorporate into the pre-trained ViT a set of prompts $\mathbf{P}$ and keys $\mathbf{K}$. We denote the overall network after incorporating the prompts as $f_{\Phi,P}$.

## 3.2 MOTIVATIONS OF OUR APPROACH

Previous prompt tuning methods propose to store a set of prompts and dedicate a subset of them for learning each task. The mechanism for choosing such task-prompt pairs relies on the query-key matching where each key represents the identifier of a prompt. This is based on the similarity between a key vector and an instance's query vector. The current SOTA prompt tuning approach is CODA (Smith et al., 2023) which relies on the query-key attention mechanism in which the prompt used for $\mathbf{x}$ is computed based on the attention of the query $q(\mathbf{x})$ (i.e., the class token at the output layer of ViT) to the keys $\mathbf{K}_1, ..., \mathbf{K}_M$. More specifically, the attention weights are computed as

$$\boldsymbol{\alpha} = (\alpha_1, ..., \alpha_M) \text{ with } \alpha_i = \gamma(q(\mathbf{x}) \otimes \mathbf{A}_i, \mathbf{K}_i) \tag{1}$$

where $\mathbf{K} = \mathbf{K}_{1:M} \in \mathbb{R}^{D \times M}$ contains keys $\mathbf{K}_{1:M}$ corresponding to the prompts $\mathbf{P} = \mathbf{P}_{1:M} \in \mathbb{R}^{L \times M}$ respectively, $\gamma(\cdot, \cdot)$ is a function to evaluate the similarity of two vectors, $\otimes$ is the element-wise product, and $\mathbf{A}_i$ is a (learnable) mask vector.

The prompt for $\mathbf{x}$ denoted by $\mathbf{P}_\mathbf{x}$ is the weighted sum of the prompts $\mathbf{P}_{1:M}$ w.r.t. the attention weights $\boldsymbol{\alpha}$ and computed as $\mathbf{P}_\mathbf{x} = \sum_i \alpha_i \mathbf{P}_i$. Eventually, the prompt $\mathbf{P}_\mathbf{x}$ is employed to derive the embedding $f_{\Phi,P}(\mathbf{x})$, subsequently used for prediction by the classification head $h_\theta$.

When fixing a set $\mathbf{P}$ of prompts for all tasks, the prompts highly used in the previous tasks are likely to be overwritten in the later tasks, hence increasingly causing the catastrophic forgetting. To mitigate the catastrophic forgetting, CODA expands a set of prompts $\mathbf{P}^t = \mathbf{P}^t_{1:M}$ and keys $\mathbf{K}^t = \mathbf{K}^t_{1:M}$ for a given task $t$. Specifically, when the task $t$ arrives, new keys $\mathbf{K}^t$ and prompts $\mathbf{P}^t$ are expanded and optimized, while the previous ones $\mathbf{K}^{1:t-1}$ and $\mathbf{P}^{1:t-1}$ are locked.

**The issue of mismatch and uncontrolled correlation in CODA:** Although this adaptable strategy certainly helps to reduce the catastrophic forgetting, it, in return, introduces another issue regarding the *mismatch in prompt representation between training and testing examples* for a given task. To further clarify this issue, let $\mathbf{x}^{tr}$ be a training example in task 1. When processing this $\mathbf{x}^{tr}$, the prompts and keys of task 2 forward do not exist yet, hence $q(\mathbf{x}^{tr})$ has no chance to do matching with these keys and prompts, implying that the corresponding prompt $\mathbf{P}_{\mathbf{x}^{tr}}$ mainly depends on the prompts $\mathbf{P}^1$ of the task 1, i.e., $\mathbf{P}_{\mathbf{x}^{tr}} = \sum_{i=1}^M \alpha_i^{tr} \mathbf{P}_i^1$ where $\boldsymbol{\alpha}^{tr}$ if corresponding weight vector. This weighted prompt is used during the training process of task 1. However, as doing inference for another testing example $\mathbf{x}^{te}$ in this task, $q(\mathbf{x}^{te})$ needs to match with the keys and prompts of all $T$ trained tasks. As a result, $\mathbf{P}_{\mathbf{x}^{te}}$ depends on the prompts $\mathbf{P}^{1:T}$ of all trained tasks, i.e., $\mathbf{P}_{\mathbf{x}^{te}} = \sum_{t=1}^T \sum_{i=1}^M \alpha_{ti}^{te} \mathbf{P}_i^t$. We name this issue as the *mismatch in prompt representation between training and testing examples*, which is illustrated in Figure 1.

Particularly, when $\mathbf{x}^{tr} = \mathbf{x}^{te} = \mathbf{x}$, the mismatch in prompt representation can be quantified as:

$$\mathbf{P}_\mathbf{x}^{te} - \mathbf{P}_\mathbf{x}^{tr} = \sum_{t=2}^T \sum_{i=1}^M \alpha_{ti} \mathbf{P}_i^t = \sum_{t=2}^T \sum_{i=1}^M \gamma(q(\mathbf{x}) \otimes \mathbf{A}_i^t, \mathbf{K}_i^t) \mathbf{P}_i^t, \tag{2}$$

representing the *mismatch* (shift) of the prompt representations of only one example of one past task, showing *accidentally uncontrolled correlations* w.r.t. this example.

Such a mismatch in prompt representation possibly leads to two critical and unsolved problems. First, since $\mathbf{x}^{te}$ and $\mathbf{x}^{tr}$ of the same task might activate different sets of prompts, the feature vectors $f_{\Phi,P}^e(\mathbf{x}^{te})$ are possibly shifted from those of $f_{\Phi,P}^c(\mathbf{x}^{tr})$. Here we note that $f_{\Phi,P}^c(\mathbf{x}^{tr})$ indicates the feature vectors computed at the end of the current task (e.g., the task 1 in our example), while $f_{\Phi,P}^e(\mathbf{x}^{te})$ indicates the feature vectors computed at the end of the training. We call this problem the *semantic shift between training and testing examples*. Second, $f_{\Phi,P}^e(\mathbf{x}^{te})$ might trigger wrong

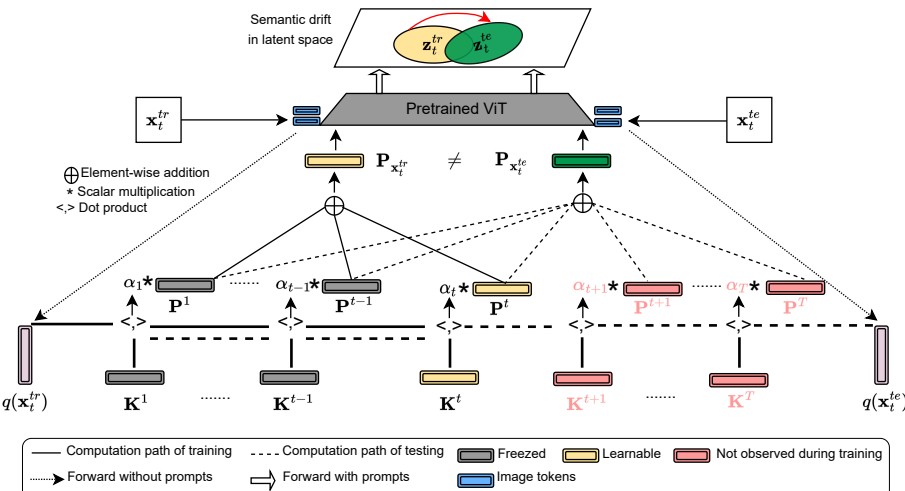

Figure 1: Visualization for the mismatch in prompt representation and semantic drift problems (Best viewed in color). When training the task $t$, each example $\mathbf{x}_t^{tr}$ of this task has a weighted prompt $\mathbf{P}_{\mathbf{x}_t^{tr}}$ that only depends on the existing prompts $\mathbf{P}^{1:t}$. Meanwhile the weighted prompt $\mathbf{P}_{\mathbf{x}_t^{te}}$ of each testing example $\mathbf{x}_t^{te}$ of the same task depends on all prompts $\mathbf{P}^{1:T}$. In case $\mathbf{x}_t^{te} = \mathbf{x}_t^{tr} = \mathbf{x}$, the weighted prompt $\mathbf{P}_{\mathbf{x}}$ gradually involves more prompts and is shifted as $T$ increases.

classification sub-heads (See Appendix A.7), leading to high prediction probabilities on this wrong classification sub-head. This is again from the fact that $\mathbf{x}^{tr}$ is trained with the weighted prompt $\mathbf{P}_{\mathbf{x}^{tr}}$ for activating the right classification sub-head of its task, while $\mathbf{x}^{te}$ is predicted using the weighted prompt $\mathbf{P}_{\mathbf{x}^{te}}$. This problem becomes even more serious for the prompt tuning approaches due to the prohibition of accessing the old data, i.e., training the classification head of the current task does not see the old data of the previous tasks, hence old data can accidentally trigger this classification sub-head.

In what follows, we present our two workarounds **(i)** *key-query orthogonal projection* to remove the mismatch issue in CODA and **(ii)** *prototype-based OVA* to address the two aforementioned problems.

## 3.3 MISMATCH AND SEMANTIC DRIFT REDUCTION BY KEY-QUERY ORTHOGONAL PROJECTION

Similar to CODA (Smith et al., 2023), we employ a task-based adaptable prompt-key system. The keys $\mathbf{K}^t = \mathbf{K}_{1:M}^t$ and prompts $\mathbf{P}^t = \mathbf{P}_{1:M}^t$ are dedicated for the task $t$. When the task $t$ arrives, we lock the old keys/prompts and only allow $\mathbf{P}^t$ and $\mathbf{K}^t$ to be updated. More specifically, given a data example $\mathbf{x}$ of the task $t$, the attention weights are computed as

$$
\begin{aligned}
\boldsymbol{\alpha} &= \gamma(q(\mathbf{x}), \mathbf{K}) \\
&= \{\gamma(q(\mathbf{x}), \mathbf{K}_{1:M}^1), \gamma(q(\mathbf{x}), \mathbf{K}_{1:M}^2), ..., \gamma(q(\mathbf{x}), \mathbf{K}_{1:M}^{t-1}), \gamma(q(\mathbf{x}), \mathbf{K}_{1:M}^t)\},
\end{aligned}
\tag{3}
$$

where $\gamma(q(\mathbf{x}), \mathbf{K}_{1:M}^i) = [\gamma(q(\mathbf{x}), \mathbf{K}_j^i)]_{j=1}^M$. Note that we do not use mask vectors as in CODA when finding the similarity between the query $q(\mathbf{x})$ and the keys, for imposing constraints on the keys more directly as shown later.

Our idea is to assure that the new $\mathbf{K}^t$ does not interfere nor correlate with $q(\mathbf{x})$ for any $\mathbf{x}$ of the tasks from 1 to $t-1$. To this end, we explicitly enforce the constraints: $q(\mathbf{x}) \perp \mathbf{K}_{1:M}^t$ or $\gamma(q(\mathbf{x}), \mathbf{K}_{1:M}^t) = \mathbf{0}$ for every $\mathbf{x}$ from every past task. By doing so, $\mathbf{P}_{\mathbf{x}}$ of a task $i$ is always dependent on the prompts $\mathbf{P}^{1:i}$ during training and testing to avoid the mismatch in prompt representation. Denote $\mathcal{S}^i$ as the subspace of query vectors from task $i \in \{1, ..., t-1\}$, and $\mathcal{Q}^{t-1} = \bigcup_{i=1}^{t-1} \mathcal{S}^i$ as the subspace spanned by query vectors from task 1 to task $t-1$. Our constraints to learn $\mathbf{K}^t$ becomes $\mathbf{K}_{1:M}^t \perp \mathcal{Q}^{t-1}$. We note that the subspace $\mathcal{S}^i$ can be computed directly using SVD and $\mathcal{Q}^{t-1}$ is updated regularly for each new task. The details of how to update $\mathcal{Q}^t$ are presented in Section A.2.

Denote $L(\mathbf{K}^{1:t-1}, \mathbf{P}^{1:t-1}, \mathbf{K}^t, \mathbf{P}^t; D^t)$ as the cross-entropy loss with respect to the training data $D^t$ at the task $t$ (i.e., $L_{CE}$). It can be simplified to $L(\mathbf{K}^t, \mathbf{P}^t; D^t)$, because $\mathbf{K}^{1:t-1}$ and $\mathbf{P}^{1:t-1}$ are locked

when learning the task $t$. Note that for the sake of simplification, we do not present the parameters of the backbone ViT and the classification heads. We need to update the keys $\mathbf{K}^t = \mathbf{K}^t_{1:M}$ such that they are orthogonal to the subspace $\mathcal{Q}^{t-1}$. A simple way is to use the idea of gradient projection memory (GPM) (Saha et al., 2021) as

$$\mathbf{K}^t = \mathbf{K}^t - \eta \nabla_{\mathbf{K}^t} L\left(\mathbf{K}^t, \mathbf{P}^t; D^t\right) \text{ and } \mathbf{K}^t_\perp = Proj_{\mathcal{Q}^{t-1}}(\mathbf{K}^t) = \mathbf{K}^t \left(\mathbf{I} - \mathbf{Q}^{t-1}\left(\mathbf{Q}^{t-1}\right)^{\mathrm{T}}\right),$$

where $\eta > 0$ is a learning rate and $\mathbf{Q}^{t-1}$ represents the matrix of the orthonormal basis of $\mathcal{Q}^{t-1}$, which is computed by concatenating the ones for the subspaces $\mathcal{S}^{1:t-1}$.

Although the above naive idea ensures $\mathbf{K}^t_\perp$ is perpendicular to the subspace $\mathcal{Q}^{t-1}$, this leads to a mismatch between $\mathbf{K}^t_\perp$ and the prompts $\mathbf{P}^t = \mathbf{P}^t_{1:M}$ which are optimized for the keys $\mathbf{K}^t$. To address this issue, we resort to the model-agnostic meta-learning (MAML) (Finn et al., 2017) to look ahead one step. Particularly, we aim to find $(\mathbf{K}^t, \mathbf{P}^t)$ such that $\mathbf{K}^t_\perp$ is compatible with $\mathbf{P}^t$ and the loss $L(\mathbf{K}^t_\perp, \mathbf{P}^t; D^t)$ is sufficiently small. This key-query orthogonal projection undertakes two phases: **(i)** *Look ahead optimization* and **(ii)** *Prompt fine tuning*, which are presented in the sequel.

**Phase 1 ("Look-ahead optimization"):** We aim to find $(\mathbf{K}^t, \mathbf{P}^t)$ in such a way that $\mathbf{K}^t_\perp$ is compatible with $\mathbf{P}^t$ and the loss $L(\mathbf{K}^t_\perp, \mathbf{P}^t; D^t)$ is sufficiently small. Therefore, we need to solve the following optimization problem:

$$\min_{\mathbf{K}^t, \mathbf{P}^t} L\left(\mathbf{K}^t_\perp, \mathbf{P}^t; D^t\right),$$

where $\mathbf{K}^t_\perp = \mathbf{K}^t \left(\mathbf{I} - \mathbf{Q}^{t-1}\left(\mathbf{Q}^{t-1}\right)^{\mathrm{T}}\right)$. Phase 1 can be concisely described as in the following flow $\mathbf{K}^t \overset{project}{\longrightarrow} \mathbf{K}^t_\perp \overset{minimize}{\longrightarrow} L\left(\mathbf{K}^t_\perp, \mathbf{P}^t; D^t\right)$.

Initially, $\mathbf{K}^t_\perp$ is computed based on $\mathbf{K}^t$ by using the GPM method. Then, this $\mathbf{K}^t_\perp$ is combined with query vectors to calculate weight vectors $\boldsymbol{\alpha}$, according to Eq. (3), which is then used to synthesize prompts for the model. Afterward the backward process requires $\mathbf{K}^t$ updated to optimize $\mathbf{K}^t_\perp$ for the objective function. In this learning process, $\mathbf{K}^t_\perp$ serves as a *"look-ahead point"* for $\mathbf{K}^t$ in which $\mathbf{K}^t_\perp$ is both guaranteed to be perpendicular to subspace $\mathcal{Q}^{t-1}$ and helps optimize the model. We undertake this look-ahead optimization in several epochs before moving to the phase 2 (cf. Algorithm 1).

**Phase 2 ("Keeping $\mathbf{K}^t \perp \mathcal{Q}^{t-1}$ fixed and then fine-tuning"):** After phase 1, the crucial step is to fix $\mathbf{K}^t = \mathbf{K}^t_\perp$. This $\mathbf{K}^t$ is unchanged and used for future inference steps. Because $\mathbf{K}^t \perp \mathcal{Q}^{t-1}$, previous query tasks will have minimal or even no interaction with these keys, and the final prompt synthesis result will not be significantly affected by $\mathbf{P}^t$, thereby eliminating the mismatch and semantic shift issues. With $\mathbf{K}^t$ is kept fixed, we proceed to fine-tune the remaining learnable components of the model to enhance their mutual alignment more effectively. We also undertake this fine-tuning phase in several epochs (cf. Algorithm 1).

## 3.4 PROTOTYPE-BASED OVA FOR BOOSTING CLASSIFICATION HEAD DISTINCTION

The key-query orthogonal projection aids to reduce the mismatch in prompt representations and the semantic drift of the training/testing examples. It also mitigates the chance that the query $q(\mathbf{x})$ of a past task $t$ uses prompts of the current/future tasks, thus the prompts $\mathbf{P}^t$ of the task $t$ have more contribution to the prompts $\mathbf{P}_\mathbf{x}$ of the example $\mathbf{x}$ of that task. Moreover, due to the potential shift in the queries $q(\mathbf{x}^{tr})$ and $q(\mathbf{x}^{te})$ for training examples $\mathbf{x}^{tr}$ and testing examples $\mathbf{x}^{te}$ in the same task and their mismatch in using the prompts as discussed in Section 3.2, the testing examples $\mathbf{x}^{te}$ might trigger wrong classification heads of a different task, thus leading to a wrong prediction.

We need a scoring mechanism to effectively strengthen the prediction probabilities of the classification head of the right task. For this purpose, we employ the one-versus-all (OVA) strategy (Saito & Saenko, 2021), which is complementary to the good task feature vector separation inherited from the key-query orthogonal projection. However, this is obstructed by the setting of prompt-tuning CL wherein the *access to old raw data* is prohibited. To bypass this obstacle, we propose prototype-based OVA for boosting the classification head distinction.

For each task $i$ from 1 to $t-1$, we employ $N$ prototypes (e.g., $N = 100$) to represent the feature vectors $f_{\Phi, P}(\mathbf{x})$, where the data examples $\mathbf{x}$ belong to task $i$ and $f_{\Phi, P}$ is the network at the end of this task. We adopt a simple strategy to conduct the prototypes by randomly selecting $N$ samples

$\mathbf{x}$ and computing the prototypes $\mathbf{p} = f_{\Phi,P}(\mathbf{x})$. Using this strategy, we obtain a compact set of prototypes across the tasks with the total size $N \times T \times d$ (e.g., $100 \times 20 \times 768$) where $T$ is the number of tasks and $d$ is the dimension of feature vectors.

**Training:** Besides $h_\theta$, let $g_\phi$ be the additional head for the OVA loss. For each task, this $g_\phi$ consists of *a single layer which returns an output of size* $2C$ ($C$ is the number of classes per task). Given an input $\mathbf{x}$, we have corresponding latent vectors $\mathbf{z} = f_{\Phi,P}(\mathbf{x})$, we also consider $\mathbf{z}$ as a prototype $\mathbf{p}$ of the task it belongs to, which will be used to train the OVA-based module without rehearsing raw data. For a specific class $c$, the output score of $\mathbf{x}$ w.r.t this class is a 2D vector $m_c(\mathbf{z}) = \text{softmax}(g_{\phi,c}(f_{\Phi,P}(\mathbf{z})))$ wherein $m_c^1(\mathbf{z}) = p(\hat{y} = c|\mathbf{z})$ specifies the in-distribution probability and $m_c^2(\mathbf{z}) = 1 - m_c^1(\mathbf{z})$ specifies the out-distribution probability. The OVA loss for one instance is defined as follows:

$$\mathcal{L}_{OVA}(\mathbf{z}, y) = -\log p(\hat{y} = y|\mathbf{z}) - \sum_{c \neq y} \log(1 - p(\hat{y} = c|\mathbf{z})). \tag{4}$$

From that, given task classification loss $L$ (our works use Cross Entropy loss $L_{CE}$ - what is often used when there is a classification head), our overall objective function is given by:

$$L = L_{CE} + \mathcal{L}_{OVA}. \tag{5}$$

**Testing:** At inference time, a sample $\mathbf{x}$ passes through the backbone $f_{\Phi,P}$, and the feature vector $\mathbf{z}$ obtained at the last layer will be passed through both the main classification head to obtain output $h_\theta(x)$ and the OVA head to obtain output $m(\mathbf{x})$. We use the OVA output scores to boost the classification head of the right task and make prediction of the task-ID of the sample $\mathbf{x}$ as follows:

(1) Extract vector $m^1(\mathbf{x}) = [m_c^1(\mathbf{x})]_c$ representing the probability of $\mathbf{x}$ belonging to class $c \in \mathcal{C}$.

(2) Calculate the score for each task based on the maximum probability of $\mathbf{x}$ belonging to that task where the task $t \in \{1, ..., T\}$ and $\mathcal{C}_t$ is the set of classes in this task:

$$score_t = \max\{m_c^1(\mathbf{x}) : c \in \mathcal{C}_t\}. \tag{6}$$

(3) Combine these task scores with output $h_\theta(\mathbf{x})$ of the main classification head. In particular, for $t \in \{1, ...T\}$ and $c \in \mathcal{C}_t$, we compute:

$$h_\theta^c(\mathbf{x}) = score_t \times h_\theta^c(\mathbf{x}). \tag{7}$$

Using the scores offered by the OVA head, we determine the probability that data point $\mathbf{x}$ belongs to a particular task, thus adjusting the output of the main head. A higher $score_t$ indicates a higher probability that x belongs to a certain class within task t, and vice versa. Eventually, we predict $\mathbf{x}$ to the class that earns *highest adjusted prediction probability* $h_\theta^c(\mathbf{x})$.

Finally, the key steps of our KOPPA are presented in Algorithm 1. More discussions about algorithm and its additional memory consumption for storing the subspace and prototypes can be found at Appendices A.1 and A.3.

## 4 EXPERIMENT

### 4.1 EXPERIMENTAL SETTINGS

**Datasets.** We use the following benchmarks:

- **Split ImageNet-R** splits 200 classes of ImageNet-R into different subsets, each of which belongs to a task. We experiment with 5-task, 10-task, and 20-task splits. This dataset is challenging because it contains diverse real-world images.
- **Split Cifar-100** is a widely used dataset in CL which similarly to Split ImageNet-R divides CIFAR100 into different tasks. We use a sequence of 10 tasks for this dataset.
- **Split DomainNet:** contains 5 tasks, each of which contains 69 classes from a total of 345 classes. The experimental results on this dataset are provided in Appendix A.6.1.

**Baselines.** We empirically compare our method with several data-free CL methods: LwF (Li & Hoiem, 2017), L2P (Wang et al., 2022c), DualPrompt (Wang et al., 2022a), CODA (Smith et al., 2023). We additionally include Joint training, which uses all data from tasks to train the model;

---

**Algorithm 1** KOPPA

---

1: Inputs: Current task data $D^t = \{(\mathbf{x^t}, y^t)\}$; Look-ahead epochs $LA_e$; Fine-tune epochs $FT_e$.
2: Initialize: New prompts and keys $\mathbf{P}^t, \mathbf{K}^t$; $\mathcal{Q}^t = \emptyset$ ; Prototype buffer $\mathcal{B} = \emptyset$
3: **if** $t = 1$ **then**
4:     1. Minimize $L_{CE}\left(\mathbf{K}^t, \mathbf{P}^t, h_\theta; D^t\right) + \mathcal{L}_{OVA}\left(g_\phi; \mathbf{z}^t, y^t\right)$     ▷ Standard training
5:     2. Compute $\mathcal{Q}^1$ and Save $\mathcal{B} = \mathcal{B} \cup \text{sample}(\mathbf{z}^1)$
6: **else**
7:     1. For e in $LA_e$:
8:         1.1. $\mathbf{K}^t_\perp = Ortho(\mathbf{K}^t, \mathcal{Q}^{t-1})$     ▷ Orthogonal look-ahead
9:         1.2. Minimize $L_{CE}\left(\mathbf{K}^t_\perp, \mathbf{P}^t, h_\theta; D^t\right) + \mathcal{L}_{OVA}\left(g_\phi; \mathbf{z}^t \cup \mathcal{B}, y^t\right)$
10:     2. Set $\mathbf{K}^t = \mathbf{K}^t_\perp$ and freeze $\overline{\mathbf{K}}^t$
11:     3. For e in $FT_e$:
12:         2.1. Minimize $L_{CE}\left(\mathbf{P}^t, h_\theta; \mathbf{K}^t, D^t\right) + \mathcal{L}_{OVA}\left(g_\phi; \mathbf{z}^t \cup \mathcal{B}, y^t\right)$
13:     4. Compute $\mathcal{Q}^t$ and save $\mathcal{B} = \mathcal{B} \cup \text{sample}(\mathbf{z}^t)$
14: **end if**

---

Fine-tuning (FT), which trains the model only on classification loss using the new task training data and its improved version - FT++, which applies the same classification head as in L2P. We also add a typical replay-based method, ER (Chaudhry et al., 2019), with a buffer of size 5,000. More details regarding the implementation of these baselines and our method are deferred to the Appendix A.5.

**Evaluation Metrics.** We report the average final accuracy ($A_N \uparrow$) and Average final Forgetting ($F_N \downarrow$), whose formal definitions can be found in the Appendix A.5. All experiments are run three times with different random seeds.

## 4.2 PERFORMANCE COMPARISON

Table 1: Average accuracy and forgetting (%) of methods across 4 benchmarks.

| Methods | S-CIFAR-100 | | S-ImageNet-R-5 | | S-ImageNet-R-10 | | S-ImageNet-R-20 | |
|---|---|---|---|---|---|---|---|---|
| | $A_N(\uparrow)$ | $F_N(\downarrow)$ | $A_N(\uparrow)$ | $F_N(\downarrow)$ | $A_N(\uparrow)$ | $F_N(\downarrow)$ | $A_N(\uparrow)$ | $F_N(\downarrow)$ |
| JOINT | 89.30 | - | 77.13 | - | 77.13 | - | 77.13 | - |
| ER | 76.20 ± 1.04 | 8.50 ± 0.37 | 71.72 ± 0.71 | 13.70 ± 0.26 | 64.43 ± 1.16 | 10.30 ± 0.05 | 52.43 ± 0.87 | 7.70 ± 0.13 |
| FT | 9.92 ± 0.27 | 29.21 ± 0.18 | 18.74 ± 0.44 | 41.49 ± 0.52 | 10.12 ± 0.51 | 25.69 ± 0.23 | 4.75 ± 0.40 | 16.34 ± 0.19 |
| FT++ | 49.91 ± 0.42 | 12.30 ± 0.23 | 60.42 ± 0.87 | 14.66 ± 0.24 | 48.93 ± 1.15 | 9.81 ± 0.31 | 35.98 ± 1.38 | 6.63 ± 0.11 |
| LwF | 64.83 ± 1.03 | 5.27 ± 0.39 | 74.56 ± 0.59 | 4.98 ± 0.37 | 66.73 ± 1.25 | 3.52 ± 0.39 | 54.05 ± 2.66 | 2.86 ± 0.26 |
| L2P++ | 82.50 ± 1.10 | 1.75 ± 0.42 | 70.83 ± 0.58 | 3.36 ± 0.18 | 69.29 ± 0.73 | 2.03 ± 0.19 | 65.89 ± 1.30 | 1.24 ± 0.14 |
| Deep L2P++ | 84.30 ± 1.03 | 1.53 ± 0.40 | 73.93 ± 0.37 | 2.69 ± 0.10 | 71.66 ± 0.64 | 1.78 ± 0.16 | 68.42 ± 1.20 | 1.12 ± 0.13 |
| DualP | 83.05 ± 1.16 | 1.72 ± 0.40 | 73.05 ± 0.50 | 2.64 ± 0.17 | 71.32 ± 0.62 | 1.71 ± 0.24 | 67.87 ± 1.39 | 1.07 ± 0.14 |
| CODA | 86.25 ± 0.74 | 1.67 ± 0.26 | 76.51 ± 0.38 | 2.99 ± 0.19 | 75.45 ± 0.56 | 1.64 ± 0.10 | 72.37 ± 1.19 | 0.96 ± 0.15 |
| KOPPA (Ours) | **97.82 ± 0.80** | **0.43 ± 0.12** | **86.02 ± 0.42** | **1.30 ± 0.12** | **91.09 ± 0.53** | **0.95 ± 0.13** | **92.89 ± 1.25** | **0.51 ± 0.15** |
| | 11.57 ↑ | | 9.51 ↑ | | 15.64 ↑ | | 20.52 ↑ | |

Foremost, we conducted a comparative analysis of our approach against established state-of-the-art methods using benchmark datasets. The outcomes presented in Table 1 clearly illustrate that KOPPA surpasses other methods based on prompt tuning across all considered cases. Our method achieves the most impressive results when exhibiting a substantial performance advantage exceeding 20% on S-Imagenet-R-20. Besides, the rehearsal-based methods, even with ample storage capacities, significantly lag behind, with the largest gap reaching up to 38.84%. Furthermore, it is noteworthy that we not only achieved superiority in terms of accuracy but also demonstrated a reduced forgetfulness rate, with the numbers being nearly halved in comparison to the leading baseline (CODA).

## 4.3 ABLATION STUDY

**Effectiveness of KOPPA in reducing semantic shift:** To illustrate the effectiveness of feature shift reduction achieved by our method, we have conducted additional experiments, as shown in Table 2, Table 3, and in Figure 2.

In Table 2, we employ Wasserstein distance to assess the feature displacement between the initial learning phase and the end of learning the final task. We denote the features acquired from $D^t$ as

Table 2: Average feature displacement from training to testing

| Task | 1 | 2 | 3 | 4 | 5 | 6 | 7 | 8 | 9 | 10 |
|---|---|---|---|---|---|---|---|---|---|---|
| CODA | **0** | 0.04 | 0.06 | 0.06 | 0.06 | 0.08 | 0.07 | 0.07 | 0.07 | 0.07 |
| KOPPA (Ours) | **0** | **0.01** | **0.01** | **0.01** | **0.01** | **0.01** | **0.01** | **0.01** | **0.01** | **0.01** |

Table 3: The effectiveness of combining prompt-based methods with our OVA-based additional head

| Methods | S-CIFAR-100 | S-ImageNet-R-5 | S-ImageNet-R-10 | S-ImageNet-R-20 |
|---|---|---|---|---|
| KOPPA (Ours) | **97.82 ± 0.80** | **86.02 ± 0.42** | **91.09 ± 1.53** | **92.89 ± 1.25** |
| CODA + OVA | 96.88 ± 0.74 | 85.32 ± 0.51 | 88.02 ± 0.65 | 92.10 ± 0.98 |

$f_{\Phi,\mathbf{P}^{1:t}}(\mathbf{x}_t)$ corresponding to the distribution $\bar{\mu}_t$ and the features after learning the last task $T$ as $f_{\Phi,\mathbf{P}^{1:T}}(\mathbf{x}_t)$ corresponding to distribution $\mu_t$. Consequently, the displacement distance is defined by $\sum_{t=1}^{T} \mathcal{W}(\bar{\mu}_t, \mu_t)$, where $\mathcal{W}$ represents the Wasserstein distance using the L2 distance. Table 2 presents the results for S-CIFAR-100 dataset. The shift observed in CODA tends to exhibit fluctuations in the early tasks before gradually stabilizing. In contrast, our method demonstrates stability from the outset, consistently indicating slighter feature shifts compared to CODA.

In addition, as discussed in the preceding section, our OVA-based approach leverages the stability of features, facilitated by our orthogonal projection learning strategy. Therefore, to provide a more comprehensive illustration of the effectiveness of our feature shift reduction, we conducted a comparative analysis, pitting our method against CODA when combined with OVA. The results in Table 3 (and Appendix A.6.2) consistently demonstrate that our approach outperforms CODA+OVA, further affirming the efficacy of our proposed method in maintaining model stability in terms of reducing feature shift.

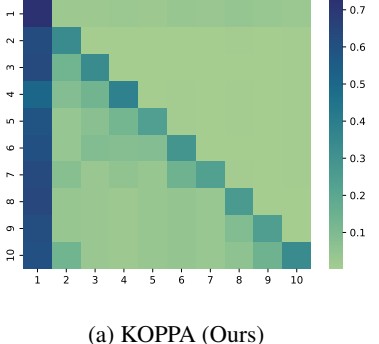
(a) KOPPA (Ours)

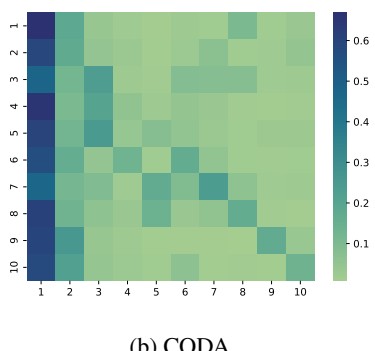
(b) CODA

Figure 2: Average weight scores of our method when doing query-key matching (S-CIFAR-100)

To further elucidate the mechanism and performance of the orthogonal projection key-query strategy, we visualize the average absolute value of the weight vector (as described in the formula 3). This visualization allows us to examine the interactions between keys and queries originating from different tasks, as depicted in Figure 2. The row index corresponds to the query's task ID, while the column index corresponds to the key's task ID. The figure reveals that positions along the main diagonal are noticeably blurred, indicating that the query from an old task has limited interaction with the key from a new task. This observation aligns with our initial intention.

**The role of the OVA head in enhancing model performance:** This section presents experimental results to clarify the role and aspects of using OVA head in our model. Table 4 (and Appendix A.6.3) present a comparison of classification methods, including our proposed approach, utilizing only traditional cross-entropy (CE) classification head, and relying solely on One-Versus-All (OVA) classification without CE. It is evident that without the integration of the OVA head, conventional classification (CE only) yields significantly lower results, with an approximate of 10% reduction in accuracy. Conversely, when employing OVA exclusively for learning, the network exhibits limited capability to acquire new knowledge and experiences notable forgetting when making erroneous

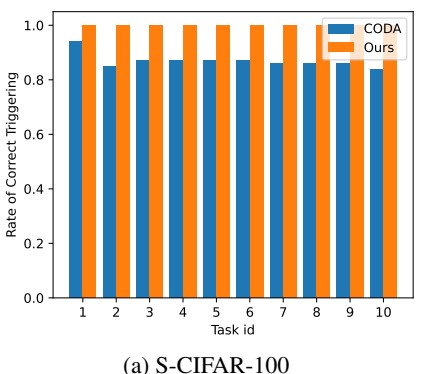

(a) S-CIFAR-100

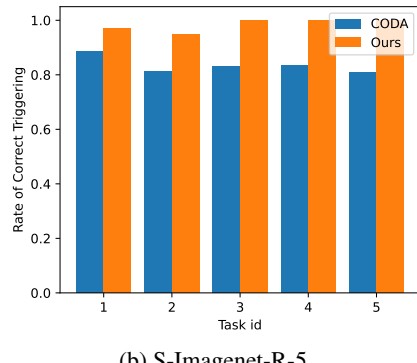

(b) S-Imagenet-R-5

Figure 3: Rate of correct triggering of KOPPA on S-CIFAR-100 and S-Imagenet-R-5.

judgments, leading to a substantial decline in accuracy. Besides, for a more comprehensive illustration of the issue discussed in the previous section regarding CODA's wrong classification head triggering and to showcase the role of OVA in addressing this concern, we present the results in Figure 3. The results highlight that the utilization of the OVA head, as in our proposed method, aids the model in selecting the appropriate classification head, leading to a substantial enhancement in final accuracy.

Furthermore, given that the training of the OVA head necessitates the storage and retrieval of prototypes from previous tasks, we investigated the effectiveness of our proposed method while varying the number of prototypes preserved for each task. The results, as presented in Table 5 (on S-CIFAR-100), indicate that, in general, an increase in the number of prototypes leads to improved performance of the OVA head. Delving deeper into the data, we observe that on the CIFAR100 and ImageNet 20T datasets, which encompass a smaller number of classes/task compared to ImageNet 5T, the model's performance exhibits a more gradual decline, demonstrating its resilience to reductions in the number of prototypes. The findings presented in Table 3 align with this observation, demonstrating that the integration of the OVA head is more beneficial in scenarios where tasks involve limited data. In this analysis, we compare the results across three data partitioning cases for different tasks originating from the same dataset (Imagenet-R).

Table 4: The effect of the losses in our KOPPA.

| Methods | S-CIFAR-100 | | S-Imagnet-R-5 | |
|---|---|---|---|---|
| | $A_N(\uparrow)$ | $F_N(\downarrow)$ | $A_N(\uparrow)$ | $F_N(\downarrow)$ |
| OVA + CE (Ours) | **97.82 ± 0.80** | **0.43 ± 0.12** | **86.02 ± 0.42** | **1.30 ± 0.12** |
| Just CE | 86.28 ± 0.81 | 1.58 ± 0.32 | 76.32 ± 0.45 | 2.85 ± 0.21 |
| Just OVA | 75.23 ± 1.13 | 2.55 ± 0.76 | 48.53 ± 1.52 | 38.59 ± 0.23 |

Table 5: Performance of our method (%) when changing the number of prototypes per task

| Dataset | 10 | 20 | 50 | 100 |
|---|---|---|---|---|
| CIFAR-100 (Ours) | 90.2 | 93.56 | 97.07 | **97.82** |
| S-ImageNet-R-5 (Ours) | 75.02 | 80.03 | 83.14 | **86.02** |
| S-ImageNet-R-20 (Ours) | 84.12 | 89.94 | 89.98 | **92.89** |

## 5    CONCLUSION

In this paper, we have investigated the potential mismatch between prompt representation of a task during training and during inference after training many more tasks, which is the result of possible correlations between keys of future tasks and queries of a past task. From this, we have proposed a novel Key-Query orthogonal projection to reduce dependence of old task queries on new task keys, thus guaranteeing old task prompt representation to be unchanged. Moreover, to enhance separation among old and new classes, we have introduced a prototype-based OVA loss, which complements the Key-Query orthogonality to obtain SOTA results in Prompt-based CL. Our analyses and extensive experiments have demonstrated the effectiveness of each of our contributions, and the significant improvement of our method.

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

# A APPENDIX

## A.1 MORE DETAILS ABOUT OUR ALGORITHM

When training a new task, data is first passed through the pretrained ViT to get the query vectors, $q(\mathbf{x}_t^{tr})$. These queries are then paired with the keys of each task learnt so far to compute their weights, $\alpha^i$, corresponding to their contribution to the current task. Specifically, the aggregated prompt for task $t$, $\mathbf{P}_{\mathbf{x}_t^{tr}}$, is the weighted sum of all prompts up to this task. The keys and prompts of old tasks are frozen and only those of the current task are learnt. To tackle the *mismatch* in prompt representation, we explicitly constrain $\mathbf{K}^t$ to be orthogonal to the subspace of queries of old tasks, $\mathcal{Q}^{t-1}$. This is achieve by our *Orthogonal Look-ahead optimization*, which seeks to simultaneously optimizes $\mathbf{P}^t$ and $\mathbf{K}^t_\perp$. Furthermore, we strengthen task classification head distinction by employing the OVA loss, $\mathcal{L}_{OVA}$, whose inputs are the prototypes of old classes and current classes. All the model parameters are learnt in an end-to-end manner.

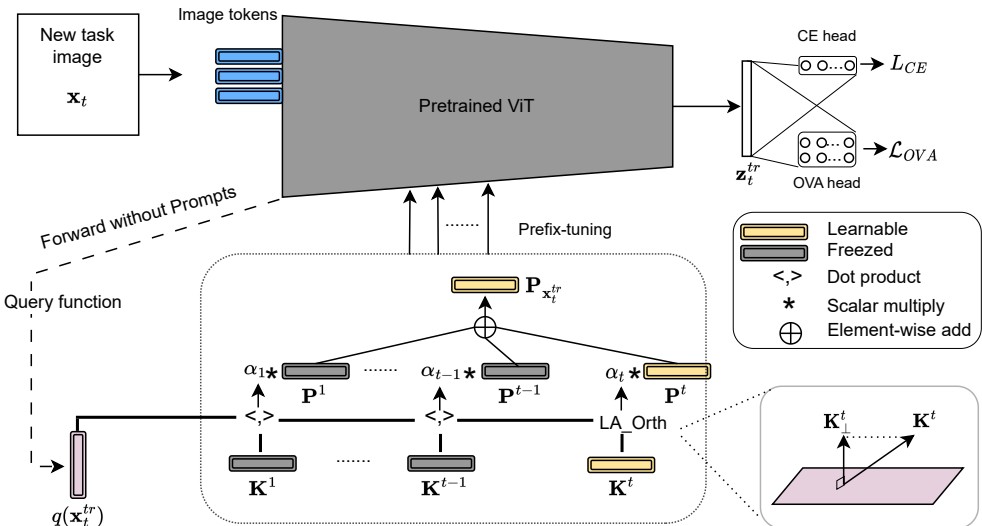

Figure 4: Training framework of KOPPA. Better viewd in color.

## A.2 HOW TO UPDATE THE MATRIX $\mathcal{Q}^t$

- In our model, we denote $\mathcal{S}^t$ as the subspace containing the query vectors of task $t$, and $\mathcal{Q}^t = \bigcup_{i=1}^t \mathcal{S}^i$ as the subspace spanned by query vectors from task 1 to the task $t$. We do not calculate $\mathcal{S}^t$ directly; instead, we obtain $\mathcal{Q}^t$ from $\mathcal{Q}^{t-1}$ as follow:

We use matrix $\mathbf{Q^t}$ to represent the subspace $\mathcal{Q}^t$, which spanned by a set of query vectors from task 1 to task $t$: $\{q(\mathbf{x_1}), q(\mathbf{x_2}), ...q(\mathbf{x_t})\}$, and this matrix is used to constraint $\mathbf{K^{t+1}}$ when learning task $t+1$. $\mathbf{Q^t}$ was computed by using SVD to choose the most representative bases. Specifically:

- For $t = 0$, let $\mathbf{R^0} = [q(\mathbf{x_1^0}), q(\mathbf{x_2^0}), ...]$ be the matrix with column vectors as query vectors of task 0. We perform SVD on $\mathbf{R^0} = \mathbf{U_0 \Sigma_0 (V_0)^T}$ and then $k$ new orthogonal bases are choosen, such that $\left\| \mathbf{R^0}_k \right\|_F^2 \geq \epsilon \left\| \mathbf{R^0} \right\|_F^2$, given the threshold $\epsilon \in (0,1)$ *(k-rank approximation)*. Finally, matrix $\mathbf{Q^0} = [u_1^0, ..., u_k^0]$ is formed with the columns being the selected basis vectors.

- For $t > 0$, let $\mathbf{R^t} = [\mathbf{Q_1^{t-1}}, \mathbf{Q_2^{t-1}}, ..., q(\mathbf{x_1^t}), q(\mathbf{x_2^t}), ...]$ where $\mathbf{Q_i^{t-1}}$ is a column vector of $\mathbf{Q^{t-1}}$. Similarly, performing SVD and k-rank approximation on $\mathbf{R^t}$, we obtain $\mathbf{Q^t}$.

In practice, in order to compute $\mathbf{Q}^t$, we use a set of 200 latent vectors $q(\mathbf{x})$ of task $t$ (and matrix $\mathbf{Q}^{t-1}$, if $t > 1$).

- Furthermore, in our implementation, the latent space has a dimension of 784. Thus, the theoretical maximum size of $\mathbf{Q}$ is $784 \times 784$ and we only need to maintain matrix $\mathbf{Q} = \mathbf{Q^{t-1}}$ to constrain $\mathbf{K^t}$ when learning task $t > 1$. In fact, the largest size of $\mathbf{Q}$ on datasets is reported in Table 6:

Table 6: Size of basis matrix $\mathbf{Q}$ after final task on datasets (KOPPA)

| Dataset | S-CIFAR-100 | S-Imagnet-R-5 | S-Imagnet-R-10 | S-Imagnet-R-20 |
|---|---|---|---|---|
| Size of $\mathbf{Q}$ | $768 \times 80$ | $768 \times 86$ | $768 \times 86$ | $768 \times 87$ |

### A.3 DISCUSSION OF ADDITIONAL MEMORY CONSUMPTION

Our KOPPA needs to store the orthonormal basis of the subspace $\mathcal{Q}^T$ corresponding to the matrix $\mathbf{Q}^T$ up to the last task $T$. This matrix has at most dimension of $768 \times 768$. In addition, for training the OVA head, we retain 100 prototypes with a dimension of 768 for each task. We view them as additional parameters to the model, which totally cost the additional memory as in Table 7.

Table 7: Additional memory that needs to store additional matrix and prototypes of KOPPA

| | S-CIFAR-100 | S-ImageNet-R-5 | S-ImageNet-R-10 | S-ImageNet-R-20 |
|---|---|---|---|---|
| Additional Memory | 4.89MB | 3.42MB | 4.89MB | 7.82MB |

For S-Imagenet-derived datasets, it is evident that our memory usage does not exceed the storage capacity equivalent to 40 images of the resolution $224 \times 224 \times 3$. Obviously, with the limited number of images for the ImageNet-R-derived datasets, achieving satisfactory results is extremely challenging. For CIFAR100, using $10 \times 100 \times 768 \times 4bytes$ prototypes equals saving 1000 raw images / 10 tasks (i.e, 100 images/tasks) to replay. We apply this raw data replay with (i) KOPPA, and (ii) just CE to update classifier head fr all tasks. The results presented in table 8 shows that replaying features is as effective as replaying raw data. Besides, replaying raw data using traditional learning methods with CE loss to finetune classifier head cannot replace the use of our OVA-based module.

| Method | $A_N(\uparrow)$ | $F_N(\downarrow)$ |
|---|---|---|
| CE + OVA (Replay $\mathbf{z}$) - Ours | **97.82 ± 0.80** | **0.43 ± 0.12** |
| CE + OVA (Replay $\mathbf{x}$) | 97.82 ± 0.80 | 0.43 ± 0.12 |
| CE (Replay $\mathbf{x}$) | 93.25 ± 0.73 | 0.85 ± 0.15 |

Table 8: Comparison between KOPPA and raw data replay solutions with the same size of buffer.

### A.4 LIMITATIONS AND POTENTIAL FUTURE WORK

About the limitations and potential future work, we would like to discuss some points as follows:

- Although this orthogonality constraint ensures (almost) no forgetting and as well as features shift, it could also hinder potential improvement of past tasks, i.e. positive backward transfer. Such cases may happen when the new task contains positively related knowledge for previous ones, hence model could learn prompts which are useful for them. Therefore, an interesting future work might be to decide when positive backward transfer may happen in the context of prompt-based CL.

- In addition, the results in Table 5 in the main paper show that the number of features used to train the OVA-module has a certain influence on the overall performance of the model. Therefore, in the future, we will consider replacing the use of raw data with effective yet flexible solutions that help the model operate more stably.

- Furthermore, when a higher level of data privacy is concerned such that no prototypes or latent features are stored, as they can be inverted to generate original images [1], our method with the OVA-based module cannot work.

## A.5 ADDITIONAL IMPLEMENTATION DETAILS

**Baselines.** In the main paper, we use LwF (Li & Hoiem, 2017), L2P (Wang et al., 2022c), Dual-Prompt (Wang et al., 2022a), CODA (Smith et al., 2023), Joint training, Fine-tuning (FT) and FT++ as data-free CIL baselines. We also add a typical replay-based method, ER (Chaudhry et al., 2019), with a buffer of size 5000.

(1) **LwF**: is a regularized-based method that enforcing the outputs of current model on new task data to be similar to those of previously trained model. This is be done using KL divergence between the two distributions outputs.
(2) **ER**: is e memory-based approach that save a fixed-size buffer of old data which is randomly selected during training or at the end of each task. When the buffer is full, reservoir sampling is adopted to decide which buffer sample will be replaced.
(3) **FT**: usually known as the lower bound for a CL method as it only trains the networking using current data without any forgetting mitigation strategy. In particular, we let FT modify classifier heads of all classes so far.
(4) **FT+**: a simple improvement of FT that only modify classifier heads of classes in the current task.
(5) **L2P++** extends L2P by using pre-fix tuning (Li & Liang, 2021), rather than prompt-tuning as in its original implementation. The first attention layer applied in this baseline. Prefix-tuning is chosen because it has beed reported to improve performance over prompt-tuning Wang et al. (2022a).
(6) **Deep L2P++** injects pre-fix tuning to four more attention layers than L2P++, resulting in a 5-layer-prefix-tuning method, which is similar to DualPrompt and CODA.

**Protocols.** We use the ImageNet-pretrained ViT-B/16 backbone (Dosovitskiy et al., 2021) for all methods. We learn model parameters with Adam optimizer, learning rate 0.001 and cosine scheduler. We implement baselines following their reported default hyper-parameters. For our method, similar to DualPrompt, we insert a component of 100 prompts with length 8 to the first 5 attention layers. We remove the attention masks in CODA, so our model has fewer trainable parameters than CODA. We train the model using 20 and 50 epochs for Split Cifar-100 and Split ImageNet-R, respectively. For the two dataset, the numbers of look-ahead epochs are 10 and 15 in that order. To effectively learn OVA head, we save a buffer size of 100 prototypes per tasks, and to find the subspace $\mathcal{Q}$, we randomly sample 200 data points of the current task to perform SVD.

**Metrics.** We use two commonly used metrics: Average Accuracy and Average Forgetting at the end of the training sequence. Specifically, denote $acc_t^i$ is the accuracy of task $t$ after the model has been trained with task $i$. Then Average Accuracy, $A_N(\uparrow)$, and Average Forgetting, $F_N(\downarrow)$, are respectively defined as:

$$A_N = \frac{1}{T}\sum_{t=1}^{T} acc_t^T, \quad F_N = \frac{1}{T-1}\sum_{i=1}^{T-1} \max_{j' \in \{1,\cdots,T-1\}} (acc_i^{j'} - acc_i^T).$$

Intuitively, a good CL model should maintain high overall results on all tasks while minimizing performance drop of previous tasks, high $A_N$ and low $F_N$. Between the two, the former is more informative in terms of balancing plasticity and stability, because a low value of accuracy $A_N$ can come with a low value of forgetting $F_N$, e.g. when the model underfits and thus forgets less.

## A.6 EXTENDED EXPERIMENTAL RESULTS

### A.6.1 RESULTS ON DOMAINNET

Table 9 presents average accuracy and forgetting of our method KOPPA and baselines on DomainNet dataset (Peng et al., 2019). Clear improvement over previous prompt-based methods can be observed with around 10 % increase in accuracy compared with the runner up, CODA-P, and with the least forgetting.

### A.6.2 MORE ON THE ROLE OF KEY ORTHOGONALITY CONSTRAINT IN REDUCING FEATURE SHIFTS

As mentioned in the main paper, the primary aim of the orthogonal projection component is to minimize feature shifts rather than pursuing a general accuracy improvement. The effectiveness of this

Table 9: Average accuracy and forgetting (%) of methods on DomainNet

| Method | $A_N(\uparrow)$ | $F_N(\downarrow)$ |
|---|---|---|
| JOINT | 79.65 | |
| ER (5000) | $58.32 \pm 0.47$ | $26.25 \pm 0.24$ |
| FT | $18.00 \pm 0.26$ | $43.55 \pm 0.27$ |
| FT++ | $39.28 \pm 0.21$ | $44.39 \pm 0.31$ |
| LwF.MC | $74.78 \pm 0.43$ | $5.01 \pm 0.14$ |
| L2P++ | $69.58 \pm 0.39$ | $2.25 \pm 0.08$ |
| Deep L2P++ | $70.54 \pm 0.51$ | $2.05 \pm 0.07$ |
| DualPrompt | $70.73 \pm 0.49$ | $2.03 \pm 0.22$ |
| CODA-P | $73.24 \pm 0.59$ | $3.46 \pm 0.09$ |
| KOPPA (Ours) | $\mathbf{84.14 \pm 0.62}$ | $\mathbf{0.23 \pm 0.10}$ |

strategy is evident in the results presented in Table 2 and Figure 2, demonstrating the superior ability to avoid shifts compared to CODA. Moreover, the outcomes in Table 10 highlight that enhanced shift avoidance plays a crucial role in enabling KOPPA to achieve superior results over CODA, particularly when combined with our proposed OVA-based module, especially on Domainnet and S-Imagenet-10 with gaps greater than 3%.

| Methods | S-CIFAR-100 | S-Imagnet-R-5 | S-Imagnet-R-10 | S-Imagnet-R-20 | DomainNet |
|---|---|---|---|---|---|
| Deep L2P++ + OVA | $95.53 \pm 0.83$ | $84.86 \pm 0.39$ | $89.23 \pm 0.77$ | $91.92 \pm 0.94$ | $80.01 \pm 0.54$ |
| DualP + OVA | $96.06 \pm 0.75$ | $85.28 \pm 0.55$ | $88.11 \pm 0.82$ | $92.13 \pm 0.84$ | $79.83 \pm 0.52$ |
| CODA + OVA | $96.88 \pm 0.74$ | $85.32 \pm 0.51$ | $88.02 \pm 0.65$ | $92.10 \pm 0.98$ | $79.76 \pm 0.55$ |
| KOPPA | $\mathbf{97.82 \pm 0.80}$ | $\mathbf{86.02 \pm 0.42}$ | $\mathbf{91.09 \pm 1.53}$ | $\mathbf{92.89 \pm 1.2}$ | $\mathbf{84.14 \pm 0.62}$ |

Table 10: Performance of other baselines when using our OVA-based module

### A.6.3 OVA VERSUS OTHER CLASSIFIERS

First, it is worth noting that OVA loss is a well-established and widely recognized loss function within the community (Rifkin & Klautau, 2004; Saito & Saenko, 2021). In this work, one of our contributions is finding out the necessity to observe all classes, then introducing a module based on the OVA loss, and how to use it to significantly improve CL performance. The high effectiveness of this module for CL should be significant for the CL literature.

Second, to illustrate the effectiveness of our proposed OVA-based module in comparison with alternative solutions, such as using a prototype-based classifier, we have conducted extended experiments. In specific:

- Prototype-based classifier (I): remove OVA head; calculate and store prototypes of classes to give predictions.
- Prototype-based classifier (II): remove OVA head, and use old prototypes and current data to learn CE head together with the backbone; calculate and store prototypes of classes to give predictions.
- CE † + CE: replace OVA head with an additional CE head and train it by using prototypes. Give predictions in the similar way that described in the main paper.

The results in Table 11 demonstrate

- (i) the essence of using a module which can observe all classes (i.e., $^*$ CE † + CE is better than Just CE);
- (ii) using just prototype to predict is a not a good option due to the indistinguishable distributions of classes, which may be the result of the uncontrolled overlapping between features from different tasks;
- (iii) our OVA module is a superior choice to all the above alternatives.

Table 11: Comparison between KOPPA's classifier vs. alternative solution. $^*$ indicates approaches that train an additional classification head using prototypes from all tasks.

| Methods | S-CIFAR-100 | S-Imagnet-R-5 | S-Imagnet-R-10 | S-Imagnet-R-20 |
|---|---|---|---|---|
| $^*$ OVA + CE (Ours) | **97.82 ± 0.80** | **86.02 ± 0.42** | **91.09 ± 0.53** | **92.89 ± 1.25** |
| Just CE | 86.28 ± 0.81 | 76.32 ± 0.45 | 75.62 ± 0.43 | 72.42 ± 1.20 |
| $^*$ CE † + CE | 94.71 ± 0.85 | 85.08 ± 0.51 | 90.02 ± 0.54 | 90.92 ± 0.88 |
| $^*$ Prototype-based classifier (I) | 67.28 ± 0.77 | 0.38 ± 0.65 | 4.75 ± 0.62 | 41.85 ± 0.92 |
| $^*$ Prototype-based classifier (II) | 69.75 ± 0.75 | 0.49 ± 0.54 | 4.91 ± 0.62 | 43.42 ± 0.85 |

### A.6.4 WHY DOES KOPPA SURPASSES JOINT?

In Tables 1 (in the main paper) and 9, it can be seen that KOPPA outperforms JOINT by a large margin. In this part, we will present in-depth reasons for this.

**First**, JOINT is better than other baselines, but we should not always consider JOINT as their upper bound.

- **Why is JOINT better than other baselines?**
  JOINT employs a training strategy where the model learns from data of all tasks simultaneously, treating them as a single task. Thus, JOINT exhibits two key advantages over other baselines: (i) no forgetting or feature shifts, and (ii) the model learns the relationships between observed classes, aligning the corresponding features effectively and minimizing misclassification. This might have been the reason why JOINT has higher results than the remaining baselines.

- **Why should we not consider JOINT as the upper bound of other baselines?**
  (i) From the code released by CODA's authors, we found that JOINT is trained on a single task, with data from all tasks. Specifically, the pre-trained model and a classification head are fine-tuned without any additional parameters such as prompts. Therefore, basically, the design of the backbone in JOINT and the other prompt-based incremental learning methods mentioned is completely different.

  (ii) Moreover, taking S-CIFAR-100 as an example, while the JOINT's model must learn to classify 100 different classes simultaneously, that of CODA's (or DualP or L2P's) only needs to learn to classify 10 classes per task. Therefore, when comparing JOINT with baselines in such a small task, JOINT may be outperformed by these baselines in terms of performance. If these methods can effectively avoid forgetting and can utilize task ID information, the final average accuracy $A_N$ can approach the average of accuracies of those small tasks which are learned independently. In other words, it is possible for JOINT to be surpassed.

**Second**, the main reasons that KOPPA surpasses JOINT by such a large margin:

*Firstly*, KOPPA's strategies enable the model to achieve the two advantages mentioned above (like JOINT) over other baselines: (i) reducing forgetting better by the proposed key-query strategy, (ii) making use of task ID information when using the OVA-based module in prediction. *Secondly*, by sequentially learning sub-tasks, KOPPA can achieve higher accuracy on each of these small tasks than JOINT (on one task of a bigger dataset). Combining the above arguments, it is convincing that KOPPA can surpass JOINT.

**Third**, we would like to provide experimental evidences. Specifically, Table 12 shows the accuracy of each task (small task) of KOPPA on S-CIFAR-100. Moreover, the corresponding triggering rate is 100% for all tasks, as shown in Figure 3 of the main paper. This implies that the final average accuracy obtained is the average of these 'sub-accuracies,' which amounts to 97.99% — higher than JOINT's 89.3%.

### A.6.5 SENSITIVITY TO PROMPT HYPER-PARAMETERS

We follow (Smith et al., 2023) to set the number of prompts used for each task and the prompt length for fair comparison. Figure 5 shows the changes of average accuracy of KOPPA and CODA when

| Task | 1 | 2 | 3 | 4 | 5 | 6 | 7 | 8 | 9 | 10 |
|---|---|---|---|---|---|---|---|---|---|---|
| Accuracy | 99.11 | 97.78 | 97.44 | 98.11 | 98.22 | 97.78 | 96.22 | 97.44 | 98.56 | 99.22 |

Table 12: Accuracy of each task, on S-CIFAR-100 (KOPPA)

the size of prompt pool and the length of prompts vary. It can be seen that KOPPA consistently outperforms CODA in all cases, and additionally, KOPPA's performance tends to increase when increasing the pool size.

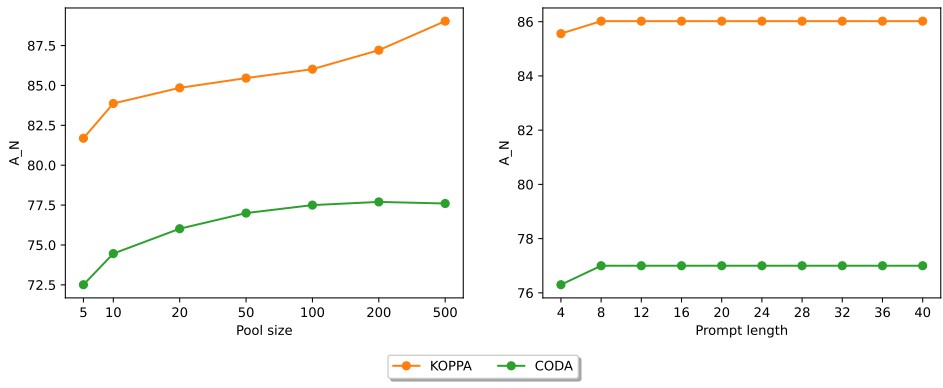

Figure 5: Average Accuracy of CODA and KOPPA with varied prompt pool sizes and prompt lengths.

### A.6.6 COMPARISON WITH HIDE AND RANPAC

Among methods using the same settings and pretrained ViT as the backbone, HiDE (Wang et al., 2023b) and RanPAC (McDonnell et al., 2023) recently stand out as the two latest state-of-the-art methods, both featuring interesting ideas and impressive results.

HiDE introduces a technique for effective representation learning, employing a contrastive regularization strategy in the form of hierarchical constraints. This approach leverages both instructed and uninstructed representations, thereby enhancing the quality of the prompt-based continual learning (CL) model. Similar to KOPPA, HiDE preserves information from old tasks through feature vector encoding to apply constraints when learning new tasks to improve prediction. However, HiDE does not address the issue of feature shifts: uninstructed representations might inadvertently select an incorrect task id. This can lead to the combination of incorrect prompts with the pre-trained backbone, resulting in uncontrolled representations, which could negatively affect the final classification quality.

Unlike HiDE and KOPPA, which are categorized under the strategy of Prompting in transformer networks, RanPAC belongs to the Class-Prototype (CP) accumulation category. RanPAC offers a comprehensive and insightful analysis of CP-based methods and introduces a solution involving a Random-Projection (RP) layer with frozen, untrained weights. This layer enhances the latent space from pre-trained models significantly. However, relying solely on the pre-trained model may hinder the model's adaptability and its ability to learn specific knowledge from new tasks, potentially limiting the method's effectiveness.

Next, we would like to provide experimental results comparing these two methods with KOPPA. The RanPAC's results are directly obtained from the original paper as we follow the same pretrained ViT architecture, while those of HiDE are reproduced using the same pretrained backbone as KOPPA and RanPAC.

The results, illustrated in Table 13, demonstrate KOPPA's superiority over these two innovative methods.

Table 13: Average Accuracy of RanPAC, HiDE and KOPPA on serveral bechmarks

| Methods | S-CIFAR-100 | S-ImageNet-R-5 | S-ImageNet-R-10 | S-ImageNet-R-20 |
|---|---|---|---|---|
| RanPAC | 92.20 | 79.90 | 77.90 | 74.50 |
| HiDE | 93.02 | 77.82 | 77.12 | 75.03 |
| KOPPA (Ours) | **97.82** | **86.02** | **91.09** | **92.89** |

### A.7 WHAT IS THE CLASSIFICATION SUB-HEAD AND THE PROBLEM OF WRONGLY TRIGGERING SUB-HEAD?

We all know that in CIL setting for classification tasks, we usually use a common prediction head. However, when we consider that the classification head can be divided into ***subheads corresponding to tasks***, then in a model where feature shift happens, there is a risk of incorrectly triggering these task-specific classification heads.

Taking the experiment on S-CIFAR100-10 as an example, we have an output of size 100 for 100 classes/10 tasks, which is returned from the common head $h_\theta$. At that time, we can consider sub-head $h_\theta^1$ for task 1, which returns the first 10 components of that output; $h_\theta^2$ for task 2, which returns the next 10 components of the output; and similarly for the remaining tasks. Assume that sub-heads $h_\theta^t$ are only learned to fit with $f_{\Phi,P}^c(\mathbf{x}^{tr})$ of sample of task $t$. Because $f_{\Phi,P}^e(\mathbf{x}^{te})$ is different from $f_{\Phi,P}^c(\mathbf{x}^{tr})$ due to feature shift so that when it interacts with these sub-heads, these heads can return uncontrollable output, leading to incorrect answers. That is, sample $\mathbf{x}$ from task $t$ could yield the highest probability prediction at the class belonging to the subhead corresponding to task $j \neq t$.

