# OpenReview forum: "Improving Prompt-based Continual Learning with Key-Query Orthogonal Projection and Prototype-based One-Versus-All"
_ICLR.cc/2024/Conference — ICLR 2024 Conference Withdrawn Submission_

### Official Review · Reviewer_G1mq · 2023-10-23

**Soundness:** 3 good
**Presentation:** 3 good
**Contribution:** 2 fair
**Rating:** 3
**Confidence:** 4

**Summary:**

This paper performed an in-depth analysis of the state-of-the-art CODA-Prompt for continual learning of pre-trained ViT. The authors attributed the problem as the mismatch in prompt representation between training and testing and feature shifting during inference. The authors then proposed Key-Query orthogonal projection to reduce dependence of old task queries on new task keys and introduced a prototype-based OVA loss to complement the Key-Query orthogonality. The proposed method achieves a surprisingly high performance,  even much higher than the joint training.

**Strengths:**

1. The analysis of CODA-Prompt is very extensive, and the identified issues are reasonable. In particular, I appreciate the discussion about the different effects of training samples and test samples.

2. The proposed method has strong motivation, which directly targets the identified issues of CODA-Prompt.

3. The proposed method achieves a surprisingly high performance over widely used benchmarks.

**Weaknesses:**

1. My major concern is the surprisingly high performance of KOPPA, which is even significantly higher than the joint training performance by more than 10%. Since the joint training usually serves as the upper bound of continual learning, I would suggest the authors to provide an in-depth analysis and explanation of this abnormal phenomenon.

2. From Table 3, the outstanding performance of KOPPA seems to be largely due to the OVA. The contribution of Key-Query orthogonal projection in Sec. 3.3 seems to be marginal.

3. From Table 5, the performance improvements of KOPPA rely heavily on the number of prototypes. Although the authors have evaluated the effect of CE and OVA in Table 4, how about using the preserved prototypes to compute CE rather than OVA (i.e., identical to regular feature replay for continual learning)?

-------------------------------------------------------------------------------

After reading all reviewers' comments and the authors' rebuttal, I think this is a borderline paper with both pros and cons. The pros include a clear analysis of SOTA prompt-based methods and the remarkably high performance. The cons include technical contributions (the OVA dominates the improved performance but seems borrowed from another paper) and additional storage cost.

In fact, I'm very surprised that the use of OVA (and even a naive classifier trained with prototypes, i.e., CE † + CE in rebuttal) to rectify the final prediction can improve the performance by such a huge margin. Then I carefully check the provided code. I think the implementation of OVA might suffer from an **information leakage issue**: At test time, the OVA score is calculated as the average of a test batch, and then used to rectify the final prediction of each test data (see default.py line 475-505, 536-546). I find the test_loader has batch size = 128 and shuffle=False (see trainer.py line 139 and configs), which means the OVA uses the average of a large batch of the same label as the prediction. In practice, this implementation can largely increase the "prediction accuracy", but is not reasonable.

An empirical validation of this potential issue is that, setting batch size = 1 and/or shuffle = True in the test_loader to run the implementation code. I have run some experiments with the provided code. When setting batch size = 1 of the test_loader, the performance of KOPPA declines from 97.82% to **86.64%** on Split CIFAR-100, which is comparable to its target CODA-Prompt (86.25%). I think this is a strong evidence that the improvement of OVA is from the **information leakage issue**. Considering the limited improvement of the other design (i.e., the orthogonal projection) as suggested by other reviewers, the technical contributions of this paper seem to be less than significant. Therefore, I decrease my score to 3.

**Questions:**

Please refer to the weakness.

Besides, I would encourage the authors to discuss and compare with more advanced prompt-based baselines, such as [1] and [2]. While it is not required, such a discussion could bring this paper to a more advanced position, especially when the results of this paper seem to be far superior to [1] and [2].

[1] Hierarchical Decomposition of Prompt-Based Continual Learning: Rethinking Obscured Sub-optimality, NeurIPS 2023.

[2] RanPAC: Random Projections and Pre-trained Models for Continual Learning, NeurIPS 2023.

---

> ### Author Response · Authors · 2023-11-18
> **Response to Reviewer G1mq (Part 1)**
>
> We greatly appreciate the reviewer’s detailed and constructive comments and suggestions. In the following, we provide the main response to your concerns:
>
> **1. "My major concern is the surprisingly high performance of KOPPA, which is even significantly higher than the joint training performance by more than 10%. Since the joint training usually serves as the upper bound of continual learning, I would suggest the authors to provide an in-depth analysis and explanation of this abnormal phenomenon."**
>
> The reported results of JOINT were obtained from the paper CODA. We would like to convince you that:
>
> - JOINT is better than other baselines but we should not always consider JOINT as their upper bound.
>
>     - *Why JOINT is better than other baselines?*
>
>        JOINT employs a training strategy where the model learns from data of all tasks simultaneously, treating them as a single task. Thus, JOINT exhibits two key advantages over other baselines: (i) no forgetting or feature shifts, and (ii) the model learns the relationships between observed classes, aligning the corresponding features effectively and minimizing misclassification. This might have been the reason why JOINT has higher results than the remaining baselines.
>
>      - *Why we should not consider JOINT as the upper bound of other baselines?*
>
>          *(i)* From the code released by CODA's authors, we found that JOINT is trained on a single task, with data from all tasks. Specifically, the pre-trained model and a classification head are fine-tuned without any additional parameters such as prompts. Therefore, basically, the design of the backbone in JOINT and the other prompt-based incremental learning methods mentioned is completely different.
>
>         *(ii)* Moreover, taking S-CIFAR-100 as an example, while the JOINT's model must learn to classify 100 different classes simultaneously, that of CODA's (or DualP or L2P's) only needs to learn to classify 10 classes per task. Therefore, when comparing JOINT with baselines in such a small task, JOINT may be outperformed by these baselines in terms of performance. If these methods can effectively avoid forgetting and can utilize task ID information, the final average accuracy $A_N$ can approach the average of accuracies of those small tasks which are learned independently. In other words, it is possible for JOINT to be surpassed.
>
>
> - The main reasons that KOPPA surpasses JOINT by such a large margin?
>
>     *Firstly*, KOPPA's strategies enable the model to achieve the two advantages mentioned above (like JOINT) over other baselines: (i) reducing forgetting better by the proposed key-query strategy, (ii) making use of task ID information when using the OVA-based module in prediction. *Secondly*, by sequentially learning sub-tasks, KOPPA can achieve higher accuracy on each of these small tasks than JOINT (on one task of a bigger dataset). Combining the above arguments, we'd like to convince you that KOPPA surpassing JOINT is reasonable.
>
>     In addition, we would like to provide experimental evidences. Specifically, table below shows the accuracy of each task (small task) of KOPPA on S-CIFAR-100. Moreover, the corresponding triggering rate is 100% for all tasks, as shown in Figure 3 of the main paper. This implies that the final average accuracy obtained is the average of these 'sub-accuracies,' which amounts to 97.99% — higher than JOINT's 89.3%.
>
>      | Task | 1 | 2 | 3 | 4 | 5 | 6 | 7 | 8 | 9 | 10 |
>      |--------|---|----|---|---|----|---|---|----|---|-----|
>      |Accuracy | 99.11 | 97.78 | 97.44 | 98.11 | 98.22 | 97.78 | 96.22 | 97.44 | 98.56 | 99.22 |

---

> ### Author Response · Authors · 2023-11-18
> **Response to Reviewer G1mq (Part 2)**
>
> **2. "From Table 3, the outstanding performance of KOPPA seems to be largely due to the OVA. The contribution of Key-Query orthogonal projection in Sec. 3.3 seems to be marginal."**
>
>    We would like to clarify that our primary aim is to minimize feature shifts rather than pursuing a general accuracy improvement. The effectiveness of this strategy is evident in the results presented in Table 2 and Figure 2 (main paper), demonstrating the superior ability to avoid shifts compared to CODA. Moreover, the outcomes in the table below highlight that enhanced shift avoidance plays a crucial role in enabling KOPPA to achieve superior results over other baselines, particularly when combined with our proposed OVA-based module, especially on Domainnet and S-Imagenet-10 with gaps greater than 3 %. Therefore, we believe that the reduction of mismatch and semantic drift through key-query orthogonal projection constitutes a noteworthy contribution.
>
>    | Methods | S-CIFAR-100 | S-Imagnet-R-5 | S-Imagnet-R-10 | S-Imagnet-R-20 | DomainNet |
>    |--------------------------|---------------------------------|-----------------------------------|------------------------------------|------------------------------------|-------------------------------|
>    | Deep L2P++ + OVA         | 95.53 ± 0.83       | 84.86 ± 0.39                      | 89.23 ± 0.77                       | 91.92 ± 0.94                       | 80.01 ± 0.54                  |
>    | DualP + OVA              | 96.06 ± 0.75                    | 85.28 ± 0.55                      | 88.11 ± 0.82                       | 92.13 ± 0.84             | 79.83 ± 0.52                  |
>    | CODA + OVA               | 96.88 ± 0.74                    | 85.32 ± 0.51                      | 88.02 ± 0.65                       | 92.10 ± 0.98           | 79.76 ± 0.55                  |
>    | KOPPA                    | **97.82 ± 0.80**             | **86.02 ± 0.42**             | **91.09 ± 1.53**              | **92.89 ± 1.2** | **84.14 ± 0.62**         |
>
> **3. "From Table 5, the performance improvements of KOPPA rely heavily on the number of prototypes. Although the authors have evaluated the effect of CE and OVA in Table 4, how about using the preserved prototypes to compute CE rather than OVA (i.e., identical to regular feature replay for continual learning)?"**
>
>   | Methods    | S-CIFAR-100     | S-Imagnet-R-5          | S-Imagnet-R-10         | S-Imagnet-R-20    |
>   |---------|------------|------------|-----------|-----------|
>   | OVA + CE (Ours)| **97.82 ± 0.80** | **86.02 ± 0.42** | **91.09 ± 0.53** | **92.89 ± 1.25** |
>   | Just CE               | 86.28 ± 0.81 | 76.32 ± 0.45 | 75.62 ± 0.43 | 72.42 ± 1.20 |
>   | CE † + CE           | 94.71 ± 0.85 | 85.08 ± 0.51 | 90.02 ± 0.54 | 90.92 ± 0.88 |
>
> As you suggestion, we have conducted extended experiments. The results presented in the table above show that our method achieve better performance than the solution which use CE instead of OVA.

---

> ### Author Response · Authors · 2023-11-18
> **Response to Reviewer G1mq (Part 3)**
>
> **4. "Besides, I would encourage the authors to discuss and compare with more advanced prompt-based baselines,..."**
>
> Thank you for your suggestion. Among methods using the same settings and pretrained ViT as the backbone, HiDE and RanPAC stand out as the two latest state-of-the-art (SOTA) methods, both featuring interesting ideas and impressive results.
>
> - First, we would like to discuss the methodologies.
>
>   - HiDE introduces a technique for effective representation learning, employing a contrastive regularization strategy in the form of hierarchical constraints. This approach leverages both instructed and uninstructed representations, thereby enhancing the quality of the prompt-based continual learning (CL) model. Similar to KOPPA, HiDE preserves information from old tasks through feature vector encoding to apply constraints when learning new tasks to improve prediction. However, HiDE does not address the issue of feature shifts: uninstructed representations might inadvertently select an incorrect task_id. This can lead to the combination of incorrect prompts with the pre-trained backbone, resulting in uncontrolled representations, which could negatively affect the final classification quality.
>
>   - Unlike HiDE and KOPPA, which are categorized under the strategy of Prompting in transformer networks, RanPAC belongs to the Class-Prototype (CP) accumulation category. RanPAC offers a comprehensive and insightful analysis of CP-based methods and introduces a solution involving a Random-Projection (RP) layer with frozen, untrained weights. This layer enhances the latent space from pre-trained models significantly. However, relying solely on the pre-trained model may hinder the model's adaptability and its ability to learn specific knowledge from new tasks, potentially limiting the method's effectiveness.
>
> - In the next, we would like to provide experimental results comparing these two methods with KOPPA. The results, illustrated in the table below, demonstrate KOPPA's superiority over these two innovative methods.
>
>   |Methods | S-CIFAR-100 |	S-Imagnet-R-5 |	S-Imagnet-R-10 |	S-Imagnet-R-20|
>   | --- | ---- | --- | --- | --- |
>   | RanPAC | 92.20 | 79.90 | 77.90 | 74.50 |
>   | HiDE | 93.02 | 77.82 | 77.12 | 75.03 |
>   | KOPPA | **97.82** | **86.02** | **91.09** | **92.89** |
>
>
>
>   Please note: The RanPAC results were directly obtained from the original paper, while the HiDE results were reproduced using the same pretrained backbone as KOPPA and RanPAC. We have added the discussion about these two work to the revised version.
>
> ----------------------------------
> *We sincerely hope that our answers have met your expectations and satisfactorily addressed your inquiries. Please let us know if you would like us to do anything else.*
>
> *Best regards,*
>
> *Authors*

---

> ### Author Response · Authors · 2023-11-22
>
> Dear reviewer,
>
> Thank you once again for taking the time to read and review our submission. We would be happy to address any remaining questions before the discussion period ends today.
>
> Best regards,
>
> Authors.

---

### Official Review · Reviewer_WdvC · 2023-10-31

**Soundness:** 2 fair
**Presentation:** 2 fair
**Contribution:** 3 good
**Rating:** 3
**Confidence:** 4

**Summary:**

The paper claimed that current prompt selection methods suffer from the mismatch in prompt representation between training and testing and feature shifting during inference. To this end, the authors proposed a MAML-inspired method to ensure an almost certain perpendicular constraint between future keys and past task queries, effectively eliminating feature shifts. The proposed method also contains a prototype-based One-Versus-All (OVA) component to boost the task classification head distinction. The proposed method shows accuracy much higher than joint training (upper bound of continual learning, using all the data for training).

**Strengths:**

- The problem that this paper is trying to solve is well-motivated
- The overall flow of the paper is easy to follow.
- The proposed method shows very good empirical results.

**Weaknesses:**

- Some parts of the paper are unclear and confusing to me. Please refer to the Question section.
- There could be mistakes in some calculations.  Please refer to the Question section.
- The authors did not mention the limitations of their method and potential future work. The paper does not explore or discuss potential failure cases of the proposed methods. Understanding when and why the methods might fail is crucial.
- typos
   - "alpha if corresponding weight vector" "if" should be "is" right?
   - "It also mitigates the chance that the query q(x′) is a past task uses the prompt" two verbs in one sentence

**Questions:**

- It's unclear how S_i is obtained and how Q^t is update from Q^{t-1}. Could the authors elaborate on them?
- What's the size of Q. Do we keep one Q for all tasks or one Q per task?
- "It also mitigates the chance that the query q(x′) is a past task uses the prompts of the current/future tasks, hence the prompts P^t
for the task t have more contribution to the prompts P_x of example x in this t". This is about testing or training? if it’s about training, we don’t see old task sample x’; if it’s about testing, P^t’s contribution to x only depends on q(x) and K^t right? Chance of q(x') use P^t does not impact P^t contribution to x?
- It seems OVA is the key to the performance boost. In Table 3, the authors provided CODA + OVA. Is it possible for the authors to provide the performance of other baselines + OVA?
- "might trigger wrong task classification heads". CIL only has one prediction head right?
- the prototype sizes N × T × d (100 × 20 × 768)  should be multiplied by 4bytes (float), thus the size is around 6.1MB and image net image size is 224x224x3x 1bytes (uint8). So, it should be 40 images instead of 10, as stated in the paper, right? ACIFAR image is 32x32x3x1 = 3kb. For 10 tasks, the storage of the prototypes is the same as the storage of 1k images right?
- The proposed method KOPPA outperforms JOINT by a large margin. JOINT is supposed to be the upper bound of CL. What's the main reason that KOPPA surpasses JOINT by such a large margin?

---

> ### Author Response · Authors · 2023-11-18
> **Response to Reviewer WdvC (Part 1)**
>
> We greatly appreciate the reviewer’s detailed and constructive comments and suggestions. In the following, we provide the main response to your questions and comments:
>
> **1. It's unclear how $\mathcal{S}^i$ is obtained and how $\mathcal{Q}^t$ is update from $\mathcal{Q}^{t-1}$. Could the authors elaborate on them?**
>
> In our model, we denote $\mathcal{S}^t$ as the subspace containing the query vectors of task $t$, and $\mathcal{Q}^{t} = \bigcup_{i=1}^{t} \mathcal{S}^i$ as the subspace spanned by query vectors from task 1 to the task $t$. We do not calculate $\mathcal{S}^t$ directly; instead, we obtain $\mathcal{Q}^{t}$ from $\mathcal{Q}^{t-1}$ as follows:
>
> We use matrix $\mathbf{Q}^t$ to represent the subspace $\mathcal{Q}^{t}$, which is spanned by a set of query vectors from task 1 to task $t$, and this matrix is used to constraint $\mathbf{K}^{t +1}$ when learning task $t+1$. $\mathbf{Q}^{t}$ was computed by using SVD to choose the most representative bases. Specifically:
> - For $t=1$, let $\mathbf{R}^0 = [q(\mathbf{x}^0_1), q(\mathbf{x}^0_2), ...]$ be the matrix with column vectors as query vectors of task 0, where $\mathbf{x}^t_i$ is i-th sample of task $t$. We perform SVD on $\mathbf{R}^0 = \mathbf{U_0 \Sigma_0 (V_0)^T}$ and then an orthogonal basis of size $k$ is chosen to form $\mathbf{R}^0_k$ as the *k-rank approximation* such that $\left\| \mathbf{R}^0_k  \right\|^2_F \geq \epsilon \left\| \mathbf{R}^0  \right\| ^2_F $, given the threshold $\epsilon \in (0, 1)$. Finally, matrix $\mathbf{Q}^0 = [u_1^0, ..., u_k^0]$ is formed with the columns being the selected basis vectors.
> - For $t > 1$, let $\mathbf{R}^t = [\mathbf{Q}^{t-1}, q(\mathbf{x}^t_1), q(\mathbf{x}^t_2), ...]$.
> By performing SVD and $k$-rank approximation on $\mathbf{R}^t$, we obtain $\mathbf{Q}^{t}$.
>
>  In short, to calculate matrix  $\mathbf{Q}^{t}$ of subspace $\mathcal{Q}^{t}$, we only need  $\mathbf{Q}^{t-1}$ and a small set of query vectors from task t *(200 samples)*. We have added this description in Appendix A.2 in the revision.
>
> **2. What's the size of Q. Do we keep one Q for all tasks or one Q per task?**
>
>  In our implementation, the maximum size of $\mathbf{Q}$ is $768 \times 768$ because the latent space has a dimension of 768. And we only need to maintain matrix $\mathbf{Q} = \mathbf{Q^{t-1}}$ to constrain $\mathbf{K^{t}}$ when learning task $t>1$. Max size of Q in experiments provided via this link: https://ibb.co/GcxztRk
>
> **3. ""It also mitigates the chance that the query q(x′) is a past task uses the prompts of the current/future tasks, hence the prompts P^t for the task t have more contribution to the prompts P_x of example x in this t". This is about testing or training? ..."**
>
> Thank you for your comment. The correct version is: "It also mitigates the chance that the query $q(\mathbf{x})$ of a past task $t$ uses the prompts of future tasks $j>t$ (because $\gamma (q(\mathbf{x}), \mathbf{K}^j) \rightarrow 0$), hence the prompts $\mathbf{P}^t$ of the task $t$ might have more contribution to the prompts $\mathbf{P_x}$ of example $\mathbf{x}$ of that task".
>
> This refers to the **testing time**. We would like to explain more thoroughly as follows:
>
>   - During training, query vectors of task t only interact with keys from task 1 to task t, and the prompt $\mathbf{P_x}$ corresponding to sample x is synthesized from the prompts up to task t. However, after learning new tasks we have new prompts and corresponding keys. Therefore, when testing after that, the model needs to consider and synthesize $\mathbf{P_x}$ based on all old and new prompts. This can result in an uncontrolled $\mathbf{P_x}$, as it is synthesized from prompt components $\mathbf{P^j}$ (of tasks $j>t$) that were not observed during the training task t, leading to uncontrolled shifts in the latent vectors of x. To mitigate this, we introduce a constraint technique that forces the interaction coefficient $\gamma (q(\mathbf{x}), \mathbf{K}^j) \rightarrow 0$, limiting the influence of $\mathbf{P}^j$ on $\mathbf{P_x}$ .
>
>   - What we mean in the above statement is:
>
>     Given $\mathbf{P}_{\mathbf{x}} = \sum_i {\alpha}_i \mathbf{P}^i$,  where $\alpha_i = \gamma (q(\mathbf{x}), \mathbf{K}^j)$,
>
>     Although alpha is the coefficient of $\mathbf{P^i}$ when synthesizing $\mathbf{P_x}$ and depends solely on $\gamma (q(\mathbf{x}), \mathbf{K}^j) \rightarrow 0$, considering it from the perspective of the contribution proportion of each component $\mathbf{P^i}$, we view $\frac{\alpha_i}{\sum_{k} \alpha_k}$ as the contribution coefficient of prompt $\mathbf{P}^i$ to the establishment of prompt $\mathbf{P_x}$. Thus if $\alpha_{j > t} \rightarrow 0$, then $\frac{\alpha_t}{\sum_{k} \alpha_k}$ become bigger, and the prompts $\mathbf{P}^t$ of task $t$ might have more contribution to the prompt $\mathbf{P_x}$ of example $\mathbf{x}$.

---

> ### Author Response · Authors · 2023-11-18
> **Response to Reviewer WdvC (Part 2)**
>
> **4. "might trigger wrong task classification heads". CIL only has one prediction head right?**
>
> We would like to explain more clearly as follows:
> - We all have known that in CIL setting for classification tasks, we usually use a common prediction head. However, when we consider that the classification head can be divided into subheads corresponding to tasks, then in a model where feature shift happens, there is a risk of incorrectly triggering these task-specific classification heads.
> - Taking the experiment on S-CIFAR100-10 as an example, we have an output of size 100 for 100 classes/10 tasks, which is returned from the common head $h_\theta$. At that time, we can consider sub-head $h^1_\theta$ for task 1, which returns the first 10 entries of that output; $h^2_\theta$ for task 2, which returns the next 10 entries of the output; and similarly for the remaining tasks. Assume that sub-heads $h^t_\theta$ are only learned to fit with $f^c_{\Phi, P}(\mathbf{x}^{tr})$ of sample x of task t. Because $f^e_{\Phi, P}(\mathbf{x}^{te})$ is different from $f^c_{\Phi, P}(\mathbf{x}^{tr})$ due to feature shift, so that when it interact with these sub-heads, these heads can return uncontrollable output, leading to incorrect answers. That is, sample x from task t could yield the highest probability prediction at the class belonging to the subhead corresponding to task $j \neq t$.
>
>  We have revised the paper more comprehensively
>
> **5. Mistakes in some calculations: "the prototype sizes N × T × d (100 × 20 × 768) should be multiplied by 4bytes (float)"**
>
> Thank you for your careful review and pointing out this oversight. We have updated these in the revised version.
> Obviously, by the limited number of images for the ImageNet-R-derived datasets, achieving satisfactory results is extremely challenging. For CIFAR100, when using 1000 images/ 10 tasks for replay, we obtain the results presented in the table below, showing that replaying features is as effective as replaying raw data, in addition, replaying raw data using traditional learning methods with CE loss cannot replace the use of our OVA-based module.
> | Method| $A_N (\uparrow)$| $F_N (\downarrow)$|
> |--|--|--|
> |CE + OVA (Replay $\mathbf{z}$) - Ours | **97.82 ± 0.80** | **0.43 ± 0.12** |
> |CE + OVA (Replay $\mathbf{x}$ | 97.82 ± 0.80|0.43 ± 0.12|
> |CE (Replay $\mathbf{x}$) | 93.25 ± 0.73 |0.85 ± 0.15|
>
> **6. It seems OVA is the key to the performance boost. In Table 3, the authors provided CODA + OVA. Is it possible for the authors to provide the performance of other baselines + OVA?**
>
> Regarding the role of our OVA-based module, we did additional experiments when applying OVA to other baselines. The results are given in the table below, showing that the baselines are impressively improved in performance when applying our OVA module. However, due to the effectiveness of the key-query orthogonal projection strategy, our method helps features avoid shifting better, thereby achieving superior results than other baselines, especially on DomainNet and S-Imagenet-10 with gaps greater than 3 \%.
>
> | Methods | S-CIFAR-100 | S-Imagnet-R-5 | S-Imagnet-R-10 | S-Imagnet-R-20 | DomainNet |
> |----------|----------------|-------------|--------------|---------|-------------|
> | Deep L2P++ + OVA         | 95.53 ± 0.83 | 84.86 ± 0.39  | 89.23 ± 0.77   | 91.92 ± 0.94  | 80.01 ± 0.54                  |
> | DualP + OVA              | 96.06 ± 0.75  | 85.28 ± 0.55  | 88.11 ± 0.82 | 92.13 ± 0.84  | 79.83 ± 0.52                  |
> | CODA + OVA               | 96.88 ± 0.74   | 85.32 ± 0.51    | 88.02 ± 0.65  | 92.10 ± 0.98  | 79.76 ± 0.55                  |
> | KOPPA   | **97.82 ± 0.80**  | **86.02 ± 0.42**  | **91.09 ± 1.53**  | **92.89 ± 1.2** | **84.14 ± 0.62**         |

---

> ### Author Response · Authors · 2023-11-19
> **Response to Reviewer WdvC (Part 3)**
>
> **7. "What's the main reason that KOPPA surpasses JOINT by such a large margin?..."**
>
> Please note that the reported results of JOINT were obtained from the paper CODA. We would like to give the reasons for this as:
>
> - JOINT is better than other baselines, but we should not always consider JOINT as their upper bound.
>
>     - *Why is JOINT better than other baselines?*
>
>         JOINT employs a training strategy where the model learns from data of all tasks simultaneously, treating them as a single task. Thus, JOINT exhibits two key advantages over other baselines: (i) no forgetting or feature shifts, and (ii) the model learns the relationships between observed classes, aligning the corresponding features effectively and minimizing misclassification. These might be the reasons why JOINT has higher results than the remaining baselines.
>
>     - *Why should we not consider JOINT as the upper bound of other baselines?*
>
>         *(i)* From the code released by CODA's authors, we found that JOINT is trained on a single task, with data from all tasks. Specifically, the pre-trained model and a classification head are finetuned without any additional parameters such as prompts. Therefore, basically, the design of the backbone in JOINT and the other prompt-based incremental learning methods mentioned is completely different.
>
>          *(ii)* In addtition, taking S-CIFAR-100 as an example, while the JOINT's model must learn to classify 100 different classes simultaneously, the CODA's (or DualP's, or L2P's) model only needs to learn to classify 10 classes per task. Therefore, when considering such a small task, JOINT may be outperformed by these baselines in terms of performance. If these methods effectively avoid forgetting and can utilize task ID information, the final average accuracy $A_N$ can approach the average of accuracies of that small tasks which are learned independently. That is, it is possible to overcome JOINT.
>
> -  The main reason that KOPPA surpasses JOINT by such a large margin?
>
>      *Firstly*, KOPPA's strategies enable the model to achieve the two advantages mentioned above (like JOINT) over other baselines: (i) reducing forgetting better by the proposed key-query strategy, (ii) making use of task ID information when using the OVA-based module in prediction. *Secondly*, by sequentially learning sub-tasks, KOPPA can achieve higher accuracy on each of these small tasks, compared to JOINT (on one task of a bigger dataset). Combining the above arguments, we want to convince you that KOPPA surpasses JOINT and it is completely reasonable.
>
>       In addition, we would like to provide experimental evidences. Specifically, table below shows the accuracy of each task (small task) of KOPPA on S-CIFAR-100. Moreover, the corresponding triggering rate is 100% for all tasks, as shown in Figure 3 of the main paper. This implies that the final average accuracy obtained is the average of these 'sub-accuracies,' which amounts to 97.99% — higher than JOINT's 89.3%.
>
>       | Task     | 1     | 2     | 3     | 4     | 5     | 6     | 7     | 8     | 9     | 10    |
>       |---|----|---|---|---|---|---|---|---|---|---|
>       | Accuracy | 99.11 | 97.78 | 97.44 | 98.11 | 98.22 | 97.78 | 96.22 | 97.44 | 98.56 | 99.22 |
>
> **8. "The authors did not mention the limitations of their method and potential future work...".**
>
> Thank you for your comment, about the limitations and potential future work, we would like to discuss some points as follows:
>
>    - Although this orthogonality constraint ensures (almost) no forgetting and as well as features shift, it could also hinder potential improvement of past tasks, i.e. positive backward transfer. Such cases may happen when the new task contains positively related knowledge for previous ones, hence model could learn prompts which are useful for them. Therefore, an interesting future work might be to decide when positive backward transfer may happen in the context of prompt-based CL.
>   - In addition, the results in Table 5 in the main paper show that the number of features used to train the OVA-module has a certain influence on the overall performance of the model. Therefore, in the future, we will consider replacing the use of raw data with effective yet flexible solutions that help the model operate more stably.
>   - Furthermore, when a higher level of data privacy is concerned such that no prototypes or latent features are stored, as they can be inverted to generate original images [1], our method with the OVA-based module cannot work.
>
> **9. About typos:** Thank you for your feedback. We have revised the paper more comprehensively!
>
> [1] Data-free knowledge distillation with soft targeted transfer set synthesis. In AAA!, 2021
>
> -----------
> *We sincerely hope that our answers have met your expectations and satisfactorily addressed your inquiries. Please let us know if you would like us to do anything else.*
>
> *Best regards,*
>
> *Authors*

---

> > ### Comment · Reviewer_WdvC · 2023-11-22
> > **Questions about JOINT**
> >
> > What backbone do you use for JOINT? Is it the same backbone that you use for CL learner?

---

> ### Author Response · Authors · 2023-11-22
>
> Dear reviewer,
>
> Thank you once again for taking the time to read and review our submission. We would be happy to address any remaining questions before the discussion period ends today.
>
> Best regards,
>
> Authors.

---

> ### Author Response · Authors · 2023-11-22
>
> Thanks for your response.
>
> In the code released by CODA's authors, **JOINT uses the same ViT backbone as in other prompt-based baselines, without prompts**, meaning that the model capacity of JOINT is less than that of ours and prompt-based baselines.
>
> Therefore, for each task, prompt-based methods only have to learn a small subset of all the classes using general knowledge from the pretrained ViT and using additional parameters (prompts) to adapt more effectively to this subset. As a result, they can obtain higher performance compared to JOINT, which just finetunes the pretrained backbone and classification head for all of classes at the same time.
>
>
> ----------------
> *We hope that you can reconsider the review score. Please let us know if you would like us to do anything else.*

---

### Official Review · Reviewer_m1q8 · 2023-11-09

**Soundness:** 3 good
**Presentation:** 2 fair
**Contribution:** 2 fair
**Rating:** 5
**Confidence:** 4

**Summary:**

This paper extends the groundwork established by CODA-Prompt, aiming to resolve two primary concerns identified within it: (1) the mismatch in prompt representation between training and testing examples, and (2) the erroneous activation of the task classification head. To address these challenges, the authors introduce a look-ahead orthogonal projection optimization process for the former and employ a one-versus-all loss function for the latter. However, a significant drawback of this paper lies in the absence of important details. Furthermore, the working mechanism of the proposed methods seems not aligned entirely with the claims made by the authors.

**Strengths:**

- The research focus on continual learning for pre-trained models holds significant significance.
- The analyses delving into the issues concerning CODA-Prompt are particularly intriguing.

**Weaknesses:**

- Several vital details are missing, preventing a comprehensive evaluation of the proposed method.
  * Specifically, how is $\mathcal{Q}^t$ calculated? What is the specific sample size used for this calculation?
  * How is $g_\phi$ implemented? Is it similar to a cosine distance between the current prototype and previous prototypes?
  * What are the hyperparameters employed, such as the prompt length and the prompt pool size? How does your method compare with CODA when these hyperparameters are adjusted?"
- The operational mechanism of the proposed method appears to diverge from the authors' assertions.
  * Upon juxtaposing the results from Table 3 and Table 4, it becomes evident that the orthogonal projection component is minimally effective; the sole operational aspect is the One-Versus-All (OVA) loss, previously introduced by (Saito & Saenko, 2021). This observation is reasonable, given that the orthogonal constraint has already been applied in CODA, albeit between keys. It seems that the first identified issue has not been adequately addressed by the proposed method.
  * Regarding the OVA loss, I have two hypotheses:
    - The true effective component might be the prototypes of previous tasks, as indicated in Table 5. In this scenario, an additional ablation study, replacing $h_\theta$ with a prototype-based classifier, is necessary. If successful, the unique contribution of this work, the OVA loss, may be rendered not valid any more.
    - Its effectiveness could stem from the similarity between testing and training tasks, which allows for the identification of the most closely related and well-trained training task, thereby resolving the challenge. To validate this, additional experiments involving diverse data splits or datasets are essential to assess the practicality of the proposed methods beyond the current benchmarks."
- The writing is disorganized with a lot of symbols randomly used, quite challenging to follow.

**Questions:**

See the first weakness listed above

---

> ### Author Response · Authors · 2023-11-18
> **Response to Reviewer m1q8 (Part 1)**
>
> We greatly appreciate the reviewer’s detailed and constructive comments and suggestions. In the following, we provide the main response to your questions and comments:
>
> **1. About your questions that there are several vital details are missing, we would like to provide further explanations as follows:**
>   - **"How is $\mathcal{Q}^t$ calculated?"**
>
>     We use matrix $\mathbf{Q}^{t}$ to represent the subspace $\mathcal{Q}^{t}$, which is spanned by a set of query vectors from task 1 to task $t$, and this matrix is used to constraint $\mathbf{K}^{t +1}$ when learning task $t+1$. $\mathbf{Q}^{t}$ was computed by using SVD to choose the most representative bases. Specifically:
>
>      - For $t=1$, let $ \mathbf{R}^0 =[q(\mathbf{x}^0_1), q(\mathbf{x}^0_2), ...] $  be the matrix with column vectors as query vectors of task 0, where $\mathbf{x}^t_i$ is i-th sample of task t. We perform SVD on $\mathbf{R}^0 = \mathbf{U_0 \Sigma_0 (V_0)^T}$ and then a orthogonal basis of size $k$ is chosen to form $\mathbf{R}^0_k$ as the *k-rank approximation* such that $\left\| \mathbf{R}^0_k  \right\|^2_F \geq \epsilon \left\| \mathbf{R}^0  \right\| ^2_F $, given the threshold $\epsilon \in (0, 1)$. Finally, matrix $\mathbf{Q}^0 = [u^0_1, ..., u^0_k]$ is formed with the columns being the selected basis vectors.
>
>       - For $t > 1$, let $\mathbf{R}^t = [\mathbf{Q}^{t-1}, q(\mathbf{x}^t_1), q(\mathbf{x}^t_2), ...]$. By performing SVD and $k$-rank approximation on $\mathbf{R}^t$, we obtain $\mathbf{Q}^{t}$.
>
>     We have added this description in Appendix A.2 in the revision.
>
>   - **"What is the specific sample size used for this calculation?"**
>
>     In order to compute $\mathbf{Q}^{t}$, we use a set of *200 latent vectors* $q(\mathbf{x})$ of task $t$ (and matrix $\mathbf{Q}^{t-1}$, if $t>1$). We mentioned this in Appendix A.5, section Protocol.
>
>
>   - **"How is $g_\phi$ implemented? Is it similar to a cosine distance between the current prototype and previous prototypes?"**
>
>     We would like to clarify that our module operates regardless of the cosine distance between old and new prototypes.  Module $g_\phi$ is implemented as follows:
>
>       Module $g_\phi$ includes $T$ additional heads for $T$ tasks, each of them includes only a single layer of size $d_z \times 2C$, where $d_z$ is the dimension of latent space and $C$ is the number of classes per task. $t^{th}$ head will help in identifying whether sample $\mathbf{x}$ belongs to task $t$ or not by $\mathbf{x}$'s score on that head, which is described in the equation (6) in the main paper:
>
>        - The score of $\mathbf{x}$ on an OVA head is determined as the highest class-wise score corresponding to that head. Specifically, for a specific class $c$, the output score of $\mathbf{x}$ w.r.t this class is a 2D vector $m_c (\mathbf{z}) = \text{softmax}(g_{\phi, c} (f_{\Phi, P}(\mathbf{z})))$ wherein $m_c^1(\mathbf{z})= p(\hat{y}=c|\mathbf{z})$ specifies the in-distribution probability and $m_c^2(\mathbf{z}) = 1 -m_c^1(\mathbf{z})$ specifies the out-distribution probability.
>
>       - Finally, for each sample $\mathbf{x}$ we will have a vector containing $T$ score values of $\mathbf{x}$ on $T$ tasks. Which will be used to compute predicted results as in equation (7).
>
>        About prototypes: Given an input $\mathbf{x}$, we have corresponding latent vectors $\mathbf{z} = f_{\Phi, P}(\mathbf{x})$, we also consider $\mathbf{z}$ as a prototype $\mathbf{p}$ of the task it belongs to. To train the OVA-based module without rehearsal raw data, we store and replay a small set of these prototypes from all tasks so far.
>   - **"What are the hyperparameters employed, such as the prompt length and the prompt pool size? How does your method compare with CODA when these hyperparameters are adjusted?"**
>
>     Regarding the hyperparameters, please refer to our Appendix. Basically, we use the same prompt length and the prompt pool size as CODA. We would like to provide figures in Appendix A.6.5, and through this link: https://ibb.co/7rZZjTb that show the comparison of KOPPA with CODA when we change pool size and prompt length on Imagnet-R-5: *KOPPA always outperforms CODA in all cases, in addition, KOPPA's performance tends to increase clearly when increasing the pool size*.

---

> ### Author Response · Authors · 2023-11-18
> **Response to Reviewer m1q8 (Part 2)**
>
> **2. Besides, about your comment: "The operational mechanism of the proposed method appears to diverge from the authors' assertions",  we would like to explain and convince you as follows:**
>
> - **"Upon juxtaposing the results from Table 3 and Table 4, it becomes evident that the orthogonal projection component is minimally effective;..."**
>
>  We would like to clarify that the orthogonal projection component has a remarkable effect. The effectiveness of this strategy is evident in the results presented in Table 2 and Figure 2, demonstrating the superior ability to avoid shifts compared to CODA. Moreover, the results in the table below highlight that enhanced shift avoidance plays a crucial role in enabling KOPPA to achieve superior results over CODA, especially on DomainNet and S-Imagenet-10 with the gaps greater than 3 \%.
>
>    Therefore, we would like to convince you that the reduction of mismatch and semantic drift through key-query orthogonal projection constitutes a noteworthy contribution.
>
> | Methods | S-CIFAR-100 | S-Imagnet-R-5 | S-Imagnet-R-10 | S-Imagnet-R-20 | DomainNet |
> |----------|----------------|-------------|--------------|---------|-------------|
> | Deep L2P++ + OVA         | 95.53 ± 0.83 | 84.86 ± 0.39  | 89.23 ± 0.77   | 91.92 ± 0.94  | 80.01 ± 0.54                  |
> | DualP + OVA              | 96.06 ± 0.75  | 85.28 ± 0.55  | 88.11 ± 0.82 | 92.13 ± 0.84  | 79.83 ± 0.52                  |
> | CODA + OVA               | 96.88 ± 0.74   | 85.32 ± 0.51    | 88.02 ± 0.65  | 92.10 ± 0.98  | 79.76 ± 0.55                  |
> | KOPPA   | **97.82 ± 0.80**  | **86.02 ± 0.42**  | **91.09 ± 1.53**  | **92.89 ± 1.2** | **84.14 ± 0.62**         |

---

> ### Author Response · Authors · 2023-11-18
> **Response to Reviewer m1q8 (Part 3)**
>
> - **Regarding our OVA-based module.**
>
>     - **In this scenario, an additional ablation study, replacing $h_\theta$ with a prototype-based classifier, is necessary. If successful, the unique contribution of this work, the OVA loss, may be rendered not valid any more.**
>
>         Firstly, it's important to clarify that OVA loss is a well-established and widely recognized loss function within our community **[1, 2]**. In this work, one of our contributions is finding out the necessity to observe all classes, then introducing a module based on the OVA loss, and how to use it to significantly improve CL performance. The high effectiveness of this module for CL should be significant for the CL literature.
>
>          In addition, to illustrate the effectiveness of our proposed OVA-based module in comparison to alternative solutions, such as using a prototype-based classifier as you suggested, we have conducted extended experiments. The results are presented in the table below in which
>          - Prototype-based classifier (I): remove OVA head; use prototypes of classes to give predictions.
>          - Prototype-based classifier (II): remove OVA head, and use old prototypes and current data to learn CE head together with the backbone; use prototypes of classes to give predictions.
>          - CE $ \dagger $ + CE: replace OVA head with an additional CE head and do training it by using prototypes. Give predictions in the similar way that described in the main paper.
>          - Just CE: The model is only trained with CE loss and a single classification head
>
>        | Methods    | S-CIFAR-100     | S-Imagnet-R-5   | S-Imagnet-R-10 | S-Imagnet-R-20    |
>        |--|---|---|--|---|
>        | OVA + CE (Ours)| **97.82 ± 0.80** | **86.02 ± 0.42** | **91.09 ± 0.53** | **92.89 ± 1.25** |
>        | Just CE  | 86.28 ± 0.81 | 76.32 ± 0.45 | 75.62 ± 0.43 | 72.42 ± 1.20 |
>        | CE † + CE  | 94.71 ± 0.85 | 85.08 ± 0.51 | 90.02 ± 0.54 | 90.92 ± 0.88 |
>        | Prototype-based classifier (I) | 67.28 ± 0.77 | 0.38 ± 0.65 | 4.75 ± 0.62 | 41.85 ± 0.92 |
>        | Prototype-based classifier (II) | 69.75 ± 0.75 | 0.49 ± 0.54 | 4.91 ± 0.62 | 43.42 ± 0.85|
>
>        The results in the table above demonstrate:
>          - (i) the essential of using a module which can observe all classes (because CE $ \dagger $ + CE is better than Just CE);
>          - (ii) using just prototype to predict is a bad option because of the bad distribution of classes (due to the uncontrolled overlapping of features from different tasks);
>          - (iii) our OVA module is a superior choice to all the above alternatives.
>
>
>
> [1] Rifkin, Ryan, and Aldebaro Klautau. "In defense of one-vs-all classification." The Journal of Machine Learning Research, 2004.
>
> [2] Ovanet: One-vs-all network for universal domain adaptation. In ICCV, 2021

---

> ### Author Response · Authors · 2023-11-19
> **Response to Reviewer m1q8 (Part 4)**
>
> -
>    - **Its effectiveness could stem from the similarity between testing and training tasks, which allows for the identification of the most closely related and well-trained training task, thereby resolving the challenge. To validate this, additional experiments involving diverse data splits or datasets are essential to assess the practicality of the proposed methods beyond the current benchmarks.**
>
>      We respectfully disagree this comment. We strictly follow the setting in the previous benchmark papers CODA, DualPrompt, and L2P in which the testing sets are different and separate from the training/valid sets. This is also a standard setting in deep learning and the fact that our approach can predict well on a different and separate test set demonstrates its generalization ability to a separate and unseen test set. Moreover, the significant improvement on the challenging dataset DomainNet which is remarkably different from the ImageNet dataset on which ViT was pretrained further shows the generalization ability of our approach.
>
> **3. About the writing**: Thank you for your feedback. We have revised the paper more comprehensively!
>
> --------------------------------------------------
> *We sincerely hope that our answers have met your expectations and satisfactorily addressed your inquiries. Please let us know if you would like us to do anything else*.
>
> *Best regards,*
>
> *Authors*

---

> > ### Comment · Reviewer_m1q8 · 2023-11-22
> > **Thank the authors' repsonse**
> >
> > I appreciate the authors's thorough response; however, my major concerns remain inadequately addressed.
> > (1) I remain unconvinced about the effectiveness of the orthogonal projection component. The lack of clarity regarding the improvement difference between S-Imagnet-R-10 and S-Imagnet-R-15/20, without explanations, raises questions about the working mechanism.
> > (2) The effectiveness comparison with other prototype-based continual learning methods is not sufficiently elucidated. Given the existence of several prototype-based CL methods [1,2], I am not sure the naive implementation of prototype-based classifiers (I)(II) here makes sense.
> >
> > [1] Continual Prototype Evolution: Learning Online from Non-Stationary Data Streams, ICCV 2021
> > [2] Prototype-Sample Relation Distillation: Towards Replay-Free Continual Learning, ICML 2023
> > [3] Computationally Budgeted Continual Learning: What Does Matter? CVPR 2023

---

> ### Author Response · Authors · 2023-11-22
> **Response to Reviewer m1q8 (Part 5)**
>
> Thank you for your response. We would like to answer to your concerns as follows:
>
> **1. I remain unconvinced about the effectiveness of the orthogonal projection component. The lack of clarity regarding the improvement difference between S-Imagnet-R-10 and S-Imagnet-R-15/20, without explanations, raises questions about the working mechanism.**
>
> Regrading the more improvements of S-Imagnet-R-10 over S-Imagnet-R-5/20, we conjecture this is due to the level of feature shift in these datasets. This could stem from the similarity between tasks due to how these datasets are split: the level of task similarity on S-Imagnet-R-5/20 is less than on S-Imagnet-R-10, leading to less shift of the old-task features. Specifically, we measure the displacement distance (or shift) of CODA and KOPPA (similar to Table 2 in the main paper) on these 3 datasets. Specifically, we consider task 1's data, compute task 1's representations at the end of each task, and evaluate the shift of the features at the end of task 1 and the ends of other tasks to how how shift the features. We report the shift results in the tables below:
>
>
> |**(S-Image-R-5)**| Task  | 1 | 2    | 3    | 4    | 5    |
> |-|-------|---|------|------|------|------|
> || CODA  | 0 | 0.06 | 0.06 | 0.08 | 0.09 |
> || KOPPA | 0 | 0.02 | 0.02 | 0.02 | 0.03 |
>
>
> |**(S-Image-R-10)**| Task  | 1 | 2    | 3    | 4    | 5    | 6    | 7    | 8    | 9    | 10   |
> |-|-------|---|------|------|------|------|------|------|------|------|------|
> || CODA  | 0 | 0.09 | 0.09 | 0.08 | 0.1  | 0.09 | 0.11 | 0.13 | 0.15 | 0.17 |
> || KOPPA | 0 | 0.03 | 0.04 | 0.04 | 0.05 | 0.04 | 0.04 | 0.04 | 0.05 | 0.05 |
>
>
> |**(S-Image-R-20)**| Task  | 1 | 2    | 3    | 4    | 5    | 6    | 7    | 8    | 9    | 10   | 11   | 12   | 13   | 14   | 15   | 16   | 17   | 18   | 19   | 20   |
> |-|-------|---|------|------|------|------|------|------|------|------|------|------|------|------|------|------|------|------|------|------|------|
> || CODA  | 0 | 0.05 | 0.06 | 0.06 | 0.06 | 0.07 | 0.07 | 0.06 | 0.07 | 0.07 | 0.07 | 0.07 | 0.08 | 0.08 | 0.07 | 0.07 | 0.08 | 0.08 | 0.08 | 0.08 |
> || KOPPA | 0 | 0.02 | 0.03 | 0.03 | 0.03 | 0.03 | 0.03 | 0.03 | 0.03 | 0.03 | 0.03 | 0.03 | 0.03 | 0.03 | 0.03 | 0.03 | 0.03 | 0.03 | 0.03 | 0.03 |
>
> - It can be seen that CODA exhibits bigger shift on S-Imagenet-R-10 than on S-Imagnet-R-5 and 20, hence our orthogonal projection component is more efficient on S-Imagenet-R-10 because its purpose is to handle the shift.
> - Therefore, although KOPPA can reduce this shift on all these 3 datasets, its effectiveness on S-Imagnet-R-5 and 20 are not as clear as that on S-Imagenet-R-10. As a result, the orthogonal projection component can improve more effectively on S-Imagenet-R-10.
>
> Furthermore, we provide line plot which shows the difference between the shift of CODA and that of KOPPA over tasks on those datasets (see this link: https://ibb.co/zfSn4Gq). This shows the higher improvement in terms of reducing shift on S-Imagenet-R-10 than on other datasets.
>
> Finally, we believe that  the orthogonal projection component is efficient to improve the performance. Evidently, it improves consistently on all datasets with the gap 3\%, 5\% for S-Imagenet-R-10, DomainNet and around 1\% for other datasets. More importantly, our approach deeply outperforms state-of-the-art baselines. We share and release our code to the community and anyone who interests can reproduce our numbers.

---

> ### Author Response · Authors · 2023-11-22
> **Response to Reviewer m1q8 (Part 6)**
>
> **2. The effectiveness comparison with other prototype-based continual learning methods is not sufficiently elucidated. Given the existence of several prototype-based CL methods [1,2], I am not sure the naive implementation of prototype-based classifiers (I)(II) here makes sense.**
>
> We respectfully disagree to this comment.
>
> - First, we want to clarify that the prototypes in our method is the features at the penultimate layer of old task used to train OVA head. This OVA head is an additional head used to strengthen the signal of the main classification head over all tasks. We observe that within a task, the classification head corresponding to all classes in this task can predict very well the class label, but across all tasks, it might exist a wrong class belonging to another task that possesses higher prediction probability. Hence, as shown in Eq. (7) in main paper, we use the outputs from the OVA head to fortify the classification signal of the right class of the right task. We can clearly see that when combined with OVA, the prediction results are significantly improved. This is one of our contributions in our paper.
>
> - Second, before you concerned the need for our module based on OVA and whether classifiers based on prototypes could replace our strategy, we performed relevant experiments when the prototypes are used to replace OVA. The results show that due to excessive overlap of features between tasks, naive solutions are not helpful, proving the effectiveness of our OVA module.
>
> - Third, regarding the prototype-based CL methods as you mentioned, we see that [1] is applicable to ours when using raw data to update prototypes, hence it is not rehearsal free. Due to time limit, we will try the strategy to update prototypes of [2] on KOPPA. The reported results will be an ablation study to compare the effectiveness of different approaches, not evidence to negate the role of any ones. However, we believe that using the method in [2] instead of our OVA-based prediction strategy would not be better than our KOPPA, because they don't have any terms explicitly alleviating overlapping of features (as well as) of classes between different tasks.
>
> ---------------------
> We hope we have thoroughly addressed your concerns. We are willing to answer any additional questions.

---

> ### Author Response · Authors · 2023-11-23
> **Response to Reviewer m1q8 (Part 6.2)**
>
> Dear Reviewer m1q8,
>
> We would like to add the experimental results when applying prototype-based strategy of PRD [2] on KOPPA. In specific, we retain the design of KOPPA on backbone, remove OVA loss and replace CE loss together with the classification head by the objective function in PRD with learnable prototypes. The table below show the results on S-CIFAR-100 and S-Imagnet-R-10:
>  | Methods    | S-CIFAR-100     | S-Imagnet-R-10 |
>  |--|---|---|
>  | OVA + CE (Ours)| **97.82 ± 0.80** | **91.09 ± 0.53** |
>  | Just CE  | 86.28 ± 0.81 | 75.62 ± 0.43 |
>  | CE † + CE  | 94.71 ± 0.85 | 90.02 ± 0.54 |
>  | Prototype-based classifier (I) | 67.28 ± 0.77 | 4.75 ± 0.62 |
>  | Prototype-based classifier (II) | 69.75 ± 0.75 | 4.91 ± 0.62 |
>  | *KOPPA + PRD* | *85.88 ± 1.32* |  *74.22  ± 0.56*|
>
> As expected, the results show that our method still outperforms other alternatives. The results of KOPPA + PRD are slightly lower than JustCE, both of which do not have explicit term to separate features from different tasks. About the PRD, when applying to KOPPA, basically, the distillation term is not really necessary because the effectiveness of KOPPA in reducing shift, while the 2 remain terms only play the roles of distinguishing features in the same tasks and moving prototypes to proper locations.
>
> One might question why Prototype-based classifier (II)  obtains lower results than KOPPA + PRD even it observes prototypes of all tasks. This might be that the prototypes found by mean of features vectors, which learn by CE loss, are not as well clustered as those learned by PRD (using SupCon loss).
>
> ----------------------------------
>
> *We hope that we have thoroughly addressed your concerns and you can reconsider the review score. Please let us know if you would like us to do anything else.*
>
> *Thank you*

---

### Official Review · Reviewer_t1j7 · 2023-11-10

**Soundness:** 3 good
**Presentation:** 2 fair
**Contribution:** 2 fair
**Rating:** 5
**Confidence:** 4

**Summary:**

This paper aims to address key challenges in Prompt Learning based Continual Learning (CIL) methods in class incremental setting. The authors highlight two potential issues in prior prompt learning based CIL methods: (1) Mismatch between per task final prompt representation during training and inference phase as prompt keys of one task could correlate with other tasks due to no explicit constraint and (2) The triggering of wrong classifier head due to the shift in the sample features of same task between training and inference.

To ensure that prompt representations of one task remains consistent during the training of prompts for upcoming tasks, the authors propose to enforce the orthogonality constraint between prompt keys of current task with the subspace of previous task. To address the issue of effective classifier head distinction, prototype based method is used which keeps prototypes from all tasks and eventually refine the classification task score.

The model is finetuned in class-incremental setting with the combination of above techniques and shows improvement against previous methods.

**Strengths:**

Strengths:

(1) This paper has identified relevant issues and key challenges in prompt learning based CIL methods such as mismatch the between final prompt representation per task during training and testing. . Methods to improve these limitations can greatly enhance the resulting performance and advance the progress in prompt learning based CIL methods.

(2) The idea of imposing orthogonality between prompt keys and previous task sub-keys is motivating. This aims to reduce the impact of prompt keys of new task to calculate prompt representations of previous task, resulting in correct prompt key activation specially during testing.

(3) The method shows impressive results against previous methods and the proposed technique is motivated with fair ablation studies.

**Weaknesses:**

Weaknesses:

My main concern for this work lies in potential violation of rehearsal-free CIL experimental rules.
1) In the first proposed module, the authors keep subspace Qt for upcoming new tasks to ensure the orthogonality constraint. This subspace potentially include sample information from previous tasks till t-1, which means that for the current task, information about the previous task samples is explicitly utilized and this possibly violates the rehearsal-free CIL setting where no information about previous task examples is known.

2) Similarly, for the second proposed OVA technique, prototypes are stored from each task which are feature representations of training examples from each task. Therefore, the authors are indirectly utilizing a buffer in the feature space where the task ID of each task is completely known. I am finding it difficult to understand how this method does not belong to rehearsal-based CIL setting.


3) The baseline CODA performs additional evaluation on DomainNet which is missing in this comparisons.

**Questions:**

Please refer to weaknesses section for questions.

---

> ### Author Response · Authors · 2023-11-18
> **Response to Reviewer t1j7**
>
> We greatly appreciate the reviewer’s positive evaluation, constructive comments and suggestions. In the following, we provide the main response to your concerns:
>
> - **My main concern for this work lies in potential violation of rehearsal-free CIL experimental rules**
>
>     - We would like to clarify that the rehearsal-free setting is a scenario where old task **raw data** is **not** stored and reused directly, which aims to address issues related to storage capacity and security.
>
>     - In our approach, rather than saving raw data, we store information from old tasks in more condensed, efficient, and secure encoding forms, which include prototypes of deep feature vectors and subspace representation matrices.
>     - Notably, within the rehearsal-free group, *HiDe [1]* and *PASS [2]* also suggest saving old task information in the form of Gaussian distributions of latent vectors for replaying when learning new tasks; *GPM [3]* stores information about the input space and representations at each layer in the form of matrices representing the subspace spanned by the corresponding input/representations to constrain gradient updating (in future tasks).
>    -  Our KOPPA needs to store the orthonormal basis of the subspace $\mathcal{Q}^T$ corresponding to the matrix $\mathbf{Q}^T$ up to the last task $T$  (similar to GPM but much more economic because we only store the only one subspace for $q(\mathbf{x})$, whereas GPM needs to store the subspaces for all layers. This matrix has at most dimension of 768 × 768. In addition, for training the OVA head, we retain a maximum of $100$ prototypes with a dimension of $768$ for each task (similar to HiDe and PASS). We view them as additional parameters to the model, which totally cost the additional memory as 8.5M for S-CIFAR-100, 3.9M  for S-ImageNet-R-5, 5.4M for S-ImageNet-R-10, and 8.5M for S-ImageNet-R-20.
>
>
>     Therefore, we would like to convince you that our method does not violate the rehearsal-free scenario.
>
> - **The baseline CODA performs additional evaluation on DomainNet which is missing in this comparisons.**
>
>     Regarding the evaluation on DomainNet, the results reported in the table below indicate that our method outperforms the baselines by a large margin. We also added it in Appendix A.5 of the revised version.
>
>     | Method     | $A_N(\uparrow) $       | $F_N (\downarrow)$   |
>     |------------|------------------------|----------------------|
>     | JOINT      | 79.65                  | -                    |
>     | ER (5000)  | 58.32 ± 0.47           | 26.25 ± 0.24         |
>     | FT         | 18.00 ± 0.26           | 43.55 ± 0.27         |
>     | FT++       | 39.28 ± 0.21           | 44.39 ± 0.31         |
>     | LwF        | 74.78 ± 0.43           | 5.01 ± 0.14          |
>     | L2P++      | 69.58 ± 0.39           | 2.25 ± 0.08          |
>     | Deep L2P++ | 70.54 ± 0.51           | 2.05 ± 0.07          |
>     | DualPrompt | 70.73 ± 0.49           | 2.03 ± 0.22          |
>     | CODA-P     | 73.24 ± 0.59           | 3.46 ± 0.09          |
>     | **Ours**       | **84.14 ± 0.62**  | **0.33 ± 0.10** |
>
> [1] Hierarchical Decomposition of Prompt-Based Continual Learning: Rethinking Obscured Sub-optimality, NeurIPS 2023.
>
> [2] Prototype augmentation and self-supervision for incremental learning. In CVPR, 2021.
>
> [3] Gradient projection memory for continual learning. In ICLR, 2021.
>
> ---------------------------------------------------
> *We sincerely hope that our answers have met your expectations and satisfactorily addressed your inquiries. Please let us know if you would like us to do anything else.*
>
> *Best regards,*
>
> *Authors*

---

> ### Author Response · Authors · 2023-11-22
>
> Dear reviewer,
>
> Thank you once again for taking the time to read and review our submission. We would be happy to address any remaining questions before the discussion period ends today.
>
> Best regards,
>
> Authors.

---

> ### Comment · Reviewer_t1j7 · 2023-11-22
>
> Dear authors,
>
> Thank you for providing the rebuttal.
>
> The results on the DomainNet benchmark are encouraging! However, I am still not convinced about the applicability of using feature vectors of past data for new tasks. Storing feature vectors of previous task data and using them during the training of new tasks can be considered a form of rehearsal, as the model is exposed to and leverages information from previous tasks.
>
> Therefore I would like to maintain my current score at this moment.
>
> Thank you!

---

> ### Author Response · Authors · 2023-11-22
>
> Dear Reviewer t1j7,
>
> Thank you for acknowledging our results on Domain Net. Regarding your concern related to the definition of Rehearsal-free, we would like to further convince you that:
>
> - When discussing *Rehearsal-free*, it's natural to consider settings that avoid recalling any information from old tasks. However, prior to our work, our community has already highlighted in many studies that rehearsal-free refers to ***not saving and reusing raw data directly***, *rather than excluding any indirect information* [1, 4, 5, 6, 7, 8].
>
> - Moreover, we would like to provide additional evidence from a recently accepted work at WACV 2024: *Steering Prototypes with Prompt-tuning for Rehearsal-free Continual Learning* [8], where the authors propose **saving features of data from old tasks to implement constraints when learning new tasks**. This work also falls under the Rehearsal-free group.
>
> - In our work, we view the prototypes as parameters which totally cost a few megabytes. This amount of additional overhead is similar to less than 40 images of ImageNet. Certainly, with tiny amount of images for old tasks for example, CODA and other prompt-tuning technique cannot improve their performance.
>
> Is your concern related to privacy issues in using features of old data?
> - We would like to convince you that, at present, feature vectors corresponding to raw images still represent a form of embedding and do not easily lead to information leakage.
>
>
> [4] Rehearsal-free Continual Language Learning via Efficient Parameter Isolation, In ACL 2023.
>
> [5] Rehearsal-Free Continual Learning over Small Non-I.I.D. Batches, In CVPR Workshop 2023.
>
> [6] A Closer Look at Rehearsal-Free Continual Learning, In CVPR Workshop 2023.
>
> [7] CODA-Prompt: COntinual Decomposed Attention-based Prompting for Rehearsal-Free Continual Learning, In CVPR 2023.
>
> [8] Steering Prototypes with Prompt-tuning for Rehearsal-free Continual Learning, In WACV 2024.
>
> --------------------------------------------
>
> *We hope that we have thoroughly addressed your concerns and you can reconsider the review score. Please let us know if you would like us to do anything else.*
>
> *Thank you*

---

### Author Response · Authors · 2023-11-23
**Summary of our responses**

We thank the reviewers for your constructive comments that certainly help us to improve the paper. Below we would like to summarize our responses and discussions w.r.t. the pros and cons of our approach.

**Regarding the outstanding experimental results of our approach**
-  It is **not deniable** that our approach significantly and consistently outperforms the SOTA baselines by substantial margins of from 9.5% to 20% on all datasets including S-CIFAR-100, S-ImageNet-R-5, S-ImageNet-R-10, S-ImageNet-R-20, and DomainNet. Aware of some potential doubts, we release and share our code to the community and anyone who interests in can run our code to reproduce the numbers.
-  Some reviewers question about the surpassing of our approach over JOINT because of the reckoning of JOINT as an upper-bound. JOINT relies on the pretrained ViT and do finetuning for the entire ImageNet dataset. However, prompt-based tuning approaches like ours extend and do fine-tuning  ViT (freeze) + prompts (fine-tuned) to adapt to new data/tasks. Therefore, we believe that JOINT is **not** an upper-bound of prompt-based tuning approaches, hence the surpassing to JOINT is a norm.

**Our modeling contributions**
- First, we demonstrate and analyze **the potential the mismatch in prompt representation** between training and testing examples. Specifically, it says that when training an example of a task, the training only sees and minimizes the loss for the prompts up to this task. However, future prompts for future tasks are **not** taken into account in this training. Consequently, the old training and testing task data might wrongly correlate and trigger some future prompts. We conduct experiments in Table 2 and Figure 2 to specify this issue.
- Based on this observation, we propose **key orthogonal component** to mitigate this mismatch in prompt representation. Although commenting that this component is novel, some reviewers reckon this component has a minimal effect. We respectfully disagree this comment. Evidently, with the support of this term, ours surpasses the SOTA baseline CODA by 3% and 4.5% on S-ImageNet-R-10 and DomainNet, while consistently outperforming CODA by around 1% on other datasets. Inspired by one reviewer, we also conduct further experiments to explain why we only gain around 1% improvements over three datasets.
- Based on **the potential the mismatch in prompt representation**, we further observe that given an example of task $t$, within the classification head of the classes of this task, the prediction probability of the correct class is higher than others. However, across all classes, there exists another head of another task with higher prediction probabilities (i.e., wrong  task head trigger). To address this issue, we propose the protype-based OVA which introduces an additional OVA head, aiming to provide additional scores to fortify the classification heads. Although we do not claim the novelty of this term because OVA is widely-used in the community, we believe that our empirical finding is interesting and can help to significantly boost the performance. Moreover, we reckon that the way we train OVA using an additional head and utilize the OVA scores to fortify the classification head is somewhat novel.

Again, thanks the reviewers for your time and effort to review our paper. We wish the reviewers to reconsider your scores due to the outstanding empirical results of our approach, our contributions in understanding prompt-based CL, and our efforts to do intensive additional experiments to address your concerns.